# Instantaneous sea ice drift speed from TanDEM-X interferometry

Dyre Oliver Dammann[1], Leif E. B. Eriksson[1], Joshua M. Jones[2], Andrew R. Mahoney[2,3], Roland Romeiser[4], Franz J. Meyer[2], Hajo Eicken[3,5], Yasushi Fukamachi[3,6,7]

[1]Department of Space, Earth, and Environment, Chalmers University of Technology, Gothenburg, Sweden
[2]Geophysical Institute, University of Alaska Fairbanks, Fairbanks, AK, USA
[3]Global Station for Arctic Research, Hokkaido University, Sapporo, Japan
[4]Rosenstiel School of Marine and Atmospheric Science, University of Miami, Miami, FL, USA
[5]International Arctic Research Center, University of Alaska Fairbanks, Fairbanks AK, USA
[6]Arctic Research Center, Hokkaido University, Sapporo, Japan
[7]Institute of Low Temperature Science, Hokkaido University, Sapporo, Japan

*Correspondence to*: Dyre Oliver Dammann (dyre.dammann@chalmers.se)

**Abstract.** The drift of sea ice is an important geophysical process with widespread implications for the ocean energy budget and ecosystems. Drifting sea ice can also threaten marine operations and present a hazard for ocean vessels and installations. Here, we evaluate single-pass along-track synthetic aperture radar (SAR) interferometry (S-ATI) as a tool to assess ice drift while discussing possible applications and inherent limitations. Initial validation shows that TanDEM-X phase-derived drift speed corresponds well with drift products from a ground-based radar at Utqiaġvik, Alaska. Joint analysis of TanDEM-X and Sentinel-1 data covering the Fram Strait demonstrates that S-ATI can help quantify the opening/closing rate of leads with possible applications for navigation. S-ATI enables an instantaneous assessment of ice drift and dynamic processes that are otherwise difficult to observe. For instance, by evaluating sea ice drift through the Vilkitsky Strait, Russia, we identified short-lived transient convergence patterns. We conclude that S-ATI enables the identification and analysis of potentially important dynamic processes (e.g. drift, rafting, and ridging). However, current limitations of S-ATI are significant (e.g. data availability and presently only provide the cross-track vector component of the ice drift field), but may be significantly reduced with future SAR systems.

## 1 Introduction

Arctic sea ice is predominately in a state of drift as a result of a near continual wind and ocean drag, which leads to redistribution and deformation. Drift processes play a large part in the sea ice thickness distribution. Differential ice motion results in the opening and closing of leads and polynyas and the formation of pressure ridges, while large-scale drift patterns control sea ice loss through export from the Arctic Ocean. Sea ice drift has therefore major implications for the mass, heat, and momentum balance of the Arctic Ocean's ice cover. Over the past several decades, Arctic sea ice has declined at a rapid rate (Stroeve et al., 2012;Comiso and Hall, 2014;Meier et al., 2014) and in confined regions resulted in more dynamic ice (Spreen et al., 2011;Kwok et al., 2013) increasing strain and fracturing (Rampal et al., 2009a). Recent and predicted changes in sea ice drift (Zhang et al., 2012) are impacting marine biota (Thomas, 2017) and coastal populations (Krupnik et al., 2010).

Sea ice drift is also a major concern for maritime activities (Eicken et al., 2009), and associated sea ice hazards play a prominent role in offshore resource development and associated coastal infrastructure (Eicken et al., 2011;Eicken and Mahoney, 2015).

The wide relevance across scientific disciplines and end users has resulted in numerous approaches for measuring ice drift. GPS buoys are an important tool to determine ice drift on pan-Arctic scales (Meier and Maslanik, 2003;Zhang et al., 2003;Rampal et al., 2009b) with unmatched temporal sampling, but are often hundreds of kilometers apart and cannot provide detailed km-scale information unless specifically deployed for validation purposes. Ground-based remote sensing systems on the other hand, such as X-band marine radars are capable of providing m-scale resolution ice drift measurements and deformation information (Druckenmiller et al., 2009;Shirasawa et al., 2013;Jones et al., 2016;Karvonen, 2016;Oikkonen et al., 2016). The coverage of ground based systems is typically limited to coastal waters, hence satellite remote sensing is also an important tool to measure ice drift. Here, microwave systems are superior due to the ability to provide information regardless of light or atmospheric conditions. Passive systems such as the Special Sensor Microwave Imager (SSM/I) are capable of providing information on the pan-Arctic scale (Kwok et al., 1998;Spreen et al., 2011) with relevance for determining sea ice age and the Arctic mass and energy budget, but with a resolution of tens of km.

Active sensors and in particular synthetic aperture radar (SAR) are capable of providing much higher resolution ice drift products at the km-scale by deriving displacement vectors between two consecutive scenes commonly through feature tracking and/or pattern matching (Hollands and Dierking, 2011;Berg and Eriksson, 2014;Karvonen, 2016;Korosov and Rampal, 2017;Muckenhuber and Sandven, 2017). These methods depend on at least two consecutive SAR scenes frequently acquired days apart. The derivation of sea ice drift speed from buoy data sampled every 1-3 days has been found to lead to underestimation of ice drift speeds by 10-20% (Haller et al., 2014) but can likely be much higher (Hutchings et al., 2011). The same bias can thus be expected when applying SAR with similar temporal sampling on the order of days. Other SAR-based approaches, such as Doppler centroid anomaly (DCA) can provide instantaneous ice drift speed, but with other inherent limitations (Kræmer et al., 2015). Instantaneous drift estimates can possibly be used to supplement traditional SAR-based ice drift algorithms as they enable evaluation of dynamics on shorter timescales. One potential application is the assessment of the ice response when impacting structures, information relevant for offshore engineering design. We here explore single-pass along-track SAR interferometry (S-ATI), which similar to DCA provides instantaneous one-dimensional drift vectors in the satellite's look direction.

InSAR is a signal processing technique which extracts the phase difference between SAR images acquired from similar viewing geometries. This interferometric phase can either signify sea ice topography if acquisitions are separated in space (i.e. non-zero perpendicular baseline) or motion in the look direction if separated in time (non-zero temporal lag). InSAR has mainly been used to study deformation (Li et al., 1996;Dammert et al., 1998;Morris et al., 1999;Vincent et al., 2004;Meyer et al., 2011;Berg et al., 2015;Marbouti et al., 2017;Dammann et al., 2018a;Dammann et al., 2018c;Dammann et al., 2019) and topography (Dammann et al., 2017;Dierking et al., 2017;Yitayew et al., 2018) of landfast sea ice since the drifting ice generally

moves too much between most satellite acquisitions to retain coherence over days to weeks. However, during the pursuit operation mode of TanDEM-X in 2010 and 2015, S-ATI analysis of drifting ice was possible with temporal lags on the order of 10 seconds (Scheiber et al., 2011;Mahoney et al., 2016;Dammann et al., 2018b). However, this lag introduces phase ambiguities that can be difficult to convert to drift speed. In contrast, we here apply bistatic acquisitions with substantially

shorter (~ 10 ms) temporal lag for the evaluation of instantaneous sea ice drift speed. This technique has been used for assessment of surface current velocity (Romeiser and Thompson, 2000;Romeiser and Runge, 2007;Romeiser et al., 2010), but to our knowledge has not been used to measure sea ice drift. In this work, we validate S-ATI measurements of sea ice drift, explore possible applications, and evaluate the limitations in particular related to data availability and the one-dimensional nature of the drift estimates.

**2 Data and methods**

**2.1 Study area and validation data**

We focus validation efforts over Utqiaġvik (formerly known as Barrow), situated in the eastern Chukchi Sea near Point Barrow, Alaska (Figure 1). We chose this region because of its diverse ice dynamics, the authors' direct experience with the region over the past two decades, and the ground-based radar stationed in Utqiaġvik continuously tracking sea ice drift. Prevailing winds

from the northeast in combination with opposing currents and the orientation of the land results in ice drift predominantly towards the southwest, persistent patches of open water, and ridged ice (Norton and Gaylord, 2004;Jones et al., 2016).

The near-shore ice out to a range of 11 km is continuously monitored using a Furuno FAR-2127 25 kW, X-band (3 cm, 10 GHz) marine radar from an altitude of 22.5 m (Figure 1) (Mahoney et al., 2015b). Radar images are archived roughly every 5-10 minutes and used for monitoring landfast ice, providing information on dynamics of offshore ice (Druckenmiller et al.,

2009). Due to occlusions and non-rigid body deformation, traditional feature tracking methods are not always effective in tracking sea ice from marine radar imagery. To reduce the noise of calculated motion vectors, we apply a combination of existing and newly developed methods. These include dense and feature-based optical flow approaches to compute motion fields from the images, active contours for delineation of stable landfast ice, and Hidden Markov Models for machine learning based event detection (Rohith et al., 2013;Jones et al., 2016). The filtering approach uses 18 consecutive images resulting in

motion products averaged over 1.5-3 hours with a grid spacing of 430 m. For features trackable over much of the radar image, errors in the dislocation vectors are on average well below 10% (Rohith et al., 2013) where an average 5 % error is attributed to uncertainties related to spatial scale and time intervals (Mahoney et al., 2015b). For the three cases evaluated here, the motion tracking algorithm only produced consistent motion vectors suitable for validation in one case (Nov 21). We also attempted validation using a 15-minute interval, which is the shortest possible interval using three images, but this resulted in

a noisy result.

To assess the ocean currents in the vicinity of our study area, we analyzed data from two moorings, M1 and M2, deployed near Utqiaġvik at 71.204N, 157.680W and 71.813N, 156.675W at a water depth of 53 m and 70 m respectively (Figure 1). The moorings contained a Teledyne RDI Workhorse Sentinel acoustic Doppler current profiler (ADCP), whose data we use to evaluate surface current velocity (Mahoney et al., 2015a) with an accuracy of better than 0.01 m s$^{-1}$ (Fukamachi et al., 2006). Velocities are derived from the Doppler shifts of return signals from particles within the water column.

In addition to Utqiaġvik, we explore the potential of InSAR-derived ice drift in the Fram and Vilkitsky Straits. Situated by the east coast of Greenland, the Fram Strait is an important location due to the dynamic conditions and large fluxes of both first- and multiyear sea ice. The Vilkitsky Strait is situated by the Taymyr Peninsula and is a strategic location on the Northern Sea Route (Council, 2009).

## 2.2 TanDEM-X data

The twin constellation TanDEM-X has operated since 2010, with a repeat-pass cycle of 11 days featuring two X-band ($\lambda$ = 3.1 cm) SAR sensors. We obtained single-look complex image pairs from the German Aerospace Center (DLR). Images were acquired in stripmap bistatic mode with short along-track baselines of less than 100 m. Image information for the images used in this work can be found in Table 1. We considered the entire data set acquired by TanDEM-X over Utqiaġvik and chose to focus on three consecutive acquisitions during 30 Oct. – 21 Nov, 2015. This time span was chosen based on (1) the ground-based radar being operational, (2) less than a kilometer wide landfast ice maximizing the ground-based radar footprint occupied by drifting ice, and (3) dynamic ice conditions. Due to increasingly later fall freeze-up of landfast ice near Utqiaġvik (Mahoney et al., 2014), sea ice is likely only a few centimeters thick during Oct. – Nov., but with potential advection of thicker ice from the eastern Beaufort Sea. The scenes were first multilooked with a resulting pixel spacing of 2.7 m and 4.7 m in range and azimuth respectively. We further followed a standard InSAR workflow (Bamler and Hartl, 1998;Ferretti et al., 2007;Dammann et al., 2016) including interferogram formation, adaptive phase filtering (Goldstein and Werner, 1998), and geocoding using the GAMMA software (Werner et al., 2000). The ice drift was derived from the interferometric phase further described in the following section. Of the three acquisitions obtained near Utqiaġvik, only the image pair from 21 Nov. was acquired when the ground-based radar was operational and could provide a coherent motion product and was thus used for validation of the derived drift.

## 2.3 InSAR-derived drift speed

The interferometric phase is represented between -$\pi$ and $\pi$. Here, only displacement in look-direction ($\Delta r_{LOS}$) results in a phase change $\Delta\Phi_{disp}$ according to $\Delta\Phi_{disp} = 4\pi \, \Delta r_{LOS}/\lambda$. With temporal baselines on the order of 10 s (i.e. TanDEM-X pursuit mode), floes can rotate or slightly deform leading to non-homogenous phase values and possibly loss of coherence. Also, for TanDEM-X, the sensor wavelength $\lambda$ is 3.1 cm, such that ice displacement exceeding $\Delta r_{LOS} \approx 1.5cm$ results in $\Delta\Phi_{disp}$ phase values to "wrap around" to the opposite side of the phase cycle causing phase ambiguities (Dammann et al., 2018b). These

ambiguities known as fringes can hide a constant phase value pertaining to the general drift speed, which was the case for Scheiber et al. (2011). In contrast, with temporal baselines on the order of 10 ms (i.e. TanDEM-X bistatic mode), each floe will feature largely homogenous phase values and typically don't wrap around since ice velocities would have to exceed roughly 1.5 m s$^{-1}$. $\Delta\Phi_{disp}$ can be converted to drift speed in the look direction using the speed of ambiguity, which is the motion

resulting in one phase cycle. We first calculate the ground range displacement in the look direction resulting in one full phase cycle (displacement of ambiguity, d), which can be expressed as:

$$d = \frac{\lambda}{2\sin\theta} \tag{1}$$

where $\theta$ is the incident angle. Furthermore, the speed of ambiguity can be expressed as $v_a = d/B_t$, where the temporal baseline $B_t = B_\parallel/v_s$, $v_s$ is the orbit speed of the satellite (7.6 km s$^{-1}$), and $B_\parallel$ is the along-track baseline (Table 1). The phase-derived

ground speed in the look direction is calculated as:

$$v_\phi = \frac{\Delta\Phi_{disp}}{2\pi} v_a \tag{2}$$

The absolute phase and motion values are initially unknown; thus, we calibrate $v_\phi$ by subtracting the derived speed of landfast ice so it is ensured to be zero. Although the direction of motion cannot be determined using the phase information alone, it is possible to determine the binary direction (i.e. whether scatterers increase or decrease their distance to the satellite) by

evaluating spatially continuous phase gradients. Here, an increasing phase is indicative of increased motion towards the satellite (if the image acquired by the leading satellite is used as a master image). We further define the positive direction such that a positive $v_\phi$ is indicative of speed towards the satellite.

$v_\phi$ is inevitably impacted by phase noise, which is introduced upon signal decorrelation. Velocity accuracy can be described:

$$\sigma_v = \frac{v_a \, \sigma_\phi}{2\pi} \tag{3}$$

Here, $\sigma_\phi$ is the standard deviation of the InSAR phase estimate, which is expressed as:

$$\sigma_\phi^2 \approx \frac{1}{2N_L} \frac{1-\gamma^2}{\gamma^2} \tag{4}$$

where N$_L$ is the independent number of looks and $\gamma$ is the interferometric coherence (Rosen et al., 2000;Dierking et al., 2017). For the data used here, the coherence is generally exceeding 0.8 leading to a velocity accuracy of $\sigma_v$~0.2 m s$^{-1}$ for unfiltered interferograms. As opposed to ice topography, motion is not expected to vary greatly for ridged floes or young ice. Therefore,

we heavily filtered the interferometric phase with an FFT window of 128 pixels. This reduces the phase noise substantially for improved accuracy likely below a few cm s$^{-1}$.

## 2.4 Drift speed ambiguities

So far, we have strictly considered phase values attributed to linear drift leading to homogenous phase values for individual floes. Additional rotational motion may need to be considered in certain cases. This will result in an along-track phase gradient across rotating floes, but was not observed in our case studies. It is also necessary to evaluate topography as a potential

contributor to phase change. The height of ambiguity, $h_a$, (i.e. the elevation that would result in one phase cycle) can be expressed:

$$h_a = \frac{\lambda R \sin \theta}{m B_\perp} \tag{5}$$

where $B_\perp$ is the perpendicular baseline, $R$ is the slant range, and $m = 1$ or 2 for monostatic and bistatic acquisitions, respectively. The potential resulting speed error, $v_e$, caused by assuming the entire phase response to be motion driven, when in reality

topographic features of height $h_0$ are present, can be determined:

$$v_e = h_o \frac{v_a}{h_a} \tag{6}$$

For the scenes considered here, $h_a$-values are roughly 40 meters hence cm-scale height offsets will result in $v_e$ on the order of mm s$^{-1}$. Therefore, the height offset between floes would have to approach one meter to make a significant contribution. This height offset would have to be prominent across an entire floe, which would reflect a difference in ice thickness and hence

would mostly be relevant for ice bergs or thick multiyear ice. Ice ridges can often feature offsets larger than a meter, but can easily be identified as a topographic response since they would otherwise indicate a non-homogenous motion across a floe, which is implausible at the timescales considered here.

In addition to topography, it is necessary to consider phase contributions from ocean waves in areas of young ice. The phase values of rigid ice floes will not be significantly impacted by waves, but fragmented ice is capable of following the vertical

motion of dm-scale wind-driven waves. The backscatter contribution from waves will often be dominated by the motion of the wave surface facing the radar. Hence, waves propagating toward the radar will result in a positive contribution to the interferometric phase due to the upward motion of the wave face seen by the radar. Conversely, waves travelling away from the radar will result in a negative phase contribution (Thompson and Jensen, 1993;Romeiser and Thompson, 2000). The speed contribution from dm-scale waves is inversely proportional to the sine of the incident angle. Hence the contribution from waves

can be substantial in cases of small incident angles and can be larger than any physical motion of the wave itself. Smaller cm-scale capillary waves can also result in a contribution to the derived speed (Valenzuela, 1978;Thompson and Jensen, 1993;Romeiser and Thompson, 2000).

# 3 Results

## 3.1 Drift speed validation near Utqiaġvik, Alaska

Three images acquired near Utqiaġvik were processed for interferometric phase and speed in the look direction and displayed in Figure 2. These acquisitions were selected based on optimal conditions, incorporating multiple drifting ice floes and narrow landfast ice extent. During the time spanned by the three acquisitions, the wind direction ranged between NNE and E predominately resulting in SW ice drift along the coast. Note that we use standard atmospheric convention for winds (by referring to the direction from which the wind is coming) and the oceanographic convention for surface currents and ice drift (by referring to the direction in which the ice or current is moving).

The image acquired on Oct. 30 features dispersed floes and open water and is displayed in Figure 2a. Areas of open water appear dark in the backscatter image due to low wind speed (~4 m s$^{-1}$) and hence low surface roughness in the form of capillary waves (see circled area in Figure 2a). The look-directional speed based on the interferometric phase is displayed in Figure 2b. Here, the exact speed has been calibrated to the known stationary ice on Elson Lagoon. Positive speed is defined as the direction opposite to look-direction (roughly towards WSW). The velocity field exhibits negative (ENE) surface velocity near the coast (see "A" in Figure 2b). This can be explained by the relatively low NNE wind speed and hence the opposing Alaska Coastal Current (Ahlnäs and Garrison, 1984;Winsor and Chapman, 2004;Jones et al., 2016) becomes the dominant force of ice drift, which at the time of the acquisition was NE at 0.4 m s$^{-1}$ as observed with mooring M1. Further off shore the speed changes orientation towards WSW, likely due to wind becoming the dominant forcing resulting in convergence around the dashed line (zero velocity) at the time of the acquisition (Figure 2b). Where positive velocity, areas of open water appears to be moving faster toward the WSW (positive direction) than the surrounding ice (top circle in Figure 2b) likely due to a wind-induced wave contribution to the phase. Here, the speed of the waves will be added to the effect of the currents resulting in apparent higher speed. Further off shore (see "B" in Figure 2b), the negative drift speed is likely due to reduced wind speed or altered wind direction since the current slowed down significantly further off shore as measured by M2 (not shown).

On Nov. 10, the wind was stronger than during the other acquisitions (10 m s$^{-1}$) and areas of open water exhibit higher backscatter than the ice floes that were present due to wind-roughening of the surface (see circled area in in Figure 2c). The strong wind results in a consistently positive drift speed in the look direction, which decreases with distance to shore (Figure 2d). This gradient was likely due to variable wind and ice forcing as the current velocity was comparable between M1 and M2. In between floes, the ocean surface exhibits velocities roughly 1 m s$^{-1}$ larger than that of the adjacent ice. This difference in speed can be explained by wind-driven waves, which are attenuated beneath the larger floes, but will have an impact over open water and looser fragmented ice. The apparent speed increase with distance from shore within the bottom circled area in Figure 2d, which would be consistent with plausible dm-scale wind-driven waves in which amplitudes increase with fetch (Walsh et al., 1989). Another reason for an increasing speed with increasing distance from the shore would be a larger concentration of frazil ice near shore reducing wave height.

At the time of acquisition on Nov. 21, the sea ice in the radar footprint consisted of a mix of large floes surrounded by young ice and open water (see circled areas in Figure 2e). It is apparent that the floes are largely drifting with homogenous speed in the southwest direction (positive speed defined as the direct opposite of look direction) (Figure 2f). The derived speed also exhibits a higher surface velocity in the areas of open water and thin ice between the floes (~1 m s$^{-1}$ vs. ~0.6 m s$^{-1}$) (circled area in Figure 2f), but less pronounced than on Nov. 10. This is likely due to reduced wind speed (7 m s$^{-1}$) and the presence of young ice in between floes (circled area in Figure 2e) damping the waves.

We further compare the TanDEM-X scene on Nov. 21 with backscatter derived from a ground-based radar system in Utqiaġvik (Figure 3a and b). Due to the high incident angle of the ground-based radar, the backscatter contrast between the ice floes and the surrounding young ice and open water is significantly greater than in the SAR imagery (see circled areas in Figure 3b).

We spatially compared the phase-derived drift speed, $v_\phi$ (figure 3c), with the drift speed derived from the ground-based radar, $v_g$ (arrows in Figure 3c). We further compared the two with $v_g$ projected into the look direction ($v_{gp}$) – the reference frame of $v_\phi$ (Figure 3d). The combined correlation has an R-value 0.86 and feature multiple outliers. The reason for this is that $v_{gp}$ is acquired over 2.5 hours and $v_\phi$ over 10 ms. This scatterplot indicates three different clusters including young ice and floes either in free drift or interacting with the landfast ice. For floes in free drift, the two datasets match within roughly 0.1 m s$^{-1}$ as a result of consistent drift speed (confirmed with the ground-based radar). However, $v_{gp}$ is generally lower, which can be explained by the 2.5 hour averaging window. $v_{gp}$ is averaged over a time period when winds fluctuated between roughly 5.5 and 7 m s$^{-1}$ while $v_\phi$ was derived when winds were in the upper range near 6.5 m s$^{-1}$ recorded in Utqiaġvik.

The correlation between $v_{gp}$ and $v_\phi$ is generally lower in areas of young ice due to the large averaging window. $v_{gp}$ in areas occupied by young ice is derived from the drift of floes occupying the respective pixels either before or after the SAR acquisition since young ice does not sustain a constant signal necessary for the feature tracking algorithm used. Therefore, the match between the derived velocities $v_\phi$ and $v_{gp}$ is poor in areas between floes (highlighted area in Figure 3d). Where floes interact with the landfast ice, drift speed is expected to be variable, which would explain outliers in Figure 3d.

To rule out a height offset between the landfast ice and the drifting ice as a possible cause for a phase offset, we calculated the drift speed error (Equation 6) to be roughly 1 mm s$^{-1}$ per cm height difference. Assuming first-year ice, a difference in freeboard between smooth sections of landfast and drifting ice greater than 5 cm is unlikely, since this would correspond to a difference in ice thickness of ~0.5 m and ice is unlikely to be thicker at this time of year. Based on local field analysis in years with particularly rough landfast ice (e.g. spring 2015), large areas of rubble ice can potentially raise the mean InSAR-derived height by 20 cm (Dammann et al., 2017). A roughness-induced height offset can often be identified through non-homogenous phase values across the rough area due to the m-scale resolution of TanDEM-X. Even so, a maximum expected offset would therefore be roughly 25 cm and lead to a 2.5 cm s$^{-1}$ height-induced bias, ruling out elevation differences as a substantial contributor to biases in the drift speed estimates.

## 3.2 Evaluating fracture dynamics near Holm Land, Greenland

From the previous section, it is clear that the interferometric phase can be used to accurately derive ice drift speed in the look direction. However, due to the calibration offsets, the absolute speed cannot be resolved without stationary landfast ice or land in the image, which can serve as a calibration point for zero drift speed. Even so, relative speed can still be resolved and is potentially of great value. One example is to determine the rate at which a lead opens or closes, which is dependent on the relative speed difference between the two sides of the lead. We applied S-ATI to two acquisitions from the Fram Strait (Figure 4a) consisting of near continuous first- and multiyear (marked "A") sea ice, which features a fracture running northwest towards Holm Land, Greenland. The main objective with this case study is to demonstrate the application to determine the opening/closing rates of fractures. In this case, it is possible to obtain absolute speed, since one of the images contains land, but this is not necessary as relative speed would be equally useful in determining opening/closing rates.

We calculated $v_\phi$ (Figure 4b) relative to the stationary ice closest to shore and define positive direction opposite to look direction. The strictly positive velocity indicates a SW velocity component. The higher speed upstream of the lead (to the NE) implies that the lead was closing at the time of the acquisition. The ice motion is not directly in response to the wind, which came from the SW at roughly 3 m s$^{-1}$, according to data from the European Center for Medium-Range Weather Forecasts' ERA5 reanalysis. To further investigate whether the fracture was in fact closing at the time of the TanDEM-X acquisition, we compared the location of the fracture edges with a Sentinel-1 image acquired 32 minutes later (Figure 5a). We delineated a section of the fracture with easily detectible boundaries in both the Sentinel-1 and TanDEM-X scenes (Figure 5b and c respectively). This comparison enabled us to estimate the closing direction (solid lines in Figure 5c) and the angle, $\theta \approx 9.2°$, relative to the TanDEM-X look direction (dashed line in Figure 5c). Comparing the fracture width in three locations (three solid lines in Figure 5c) indicated that the fracture closed by roughly 200m (176-244 m) during the 32 minutes between acquisitions. This corresponds to a closing velocity of $10.9 \pm 1.8$ cm s$^{-1}$. From Figure 4b, the difference in $v_\phi$ across this lead (along the three solid lines) is approximately $10.0 \pm 1.0$ cm s$^{-1}$, which corresponds to an instantaneous speed difference of $10.1 \pm 1.1$ cm s$^{-1}$ in the direction of lead closure. This is within ~10% of the closure rate estimated from the comparison of TanDEM-X and Sentinel-1 imagery and within the window of uncertainty. The difference could be due to variation in the closure rate over time. At these closure speeds, the 1 km wide fracture would have closed completely within approximately 3 hours. A Sentinel image acquired 15 hours later (not shown) confirms that the lead closed.

## 3.3 Assessing drift zones in Vilkitsky Strait, Russia

We further examined an additional case study in the Vilkitsky Strait, an area with relevance in the context of maritime navigation, to demonstrate the use of S-ATI in a dynamically complex scenario (not homogenous floe speeds as in previous sections). The Strait near Taimyr Peninsula features either landfast ice or temporarily stationary pack ice (as absence of landfast ice is apparent in ice charts by the Arctic and Antarctic Research Institute – www.aari.ru). Stationary pack ice is also present south of Bolshevik Island in Severnaya Zemlya visible with lower backscatter and zero ice drift in the look direction in Figure 6a and b respectively (see "A" and "B" in Figure 6b). Between these areas of stationary ice are two distinct intermediate zones

of ice moving at roughly 0.3 m s$^{-1}$ ("C" and "D") bordering a channel with higher velocities ranging between roughly 0.35 and 0.45 m s$^{-1}$ ("E-H"). The drift is approximately eastward in response to a WSW wind of roughly 6 m s$^{-1}$ obtained from ERA5 reanalysis. The westerly wind leads to open water on the east side adjacent to the peninsula (see "I" in Figure 6a), allowing otherwise confined ice to move more freely leading to larger velocities ("E"). The central channel ("F"-"H") exhibits higher

drift speed than the ice immediately to each side ("C" and "D") and variable speed in the form of a ramping speed ("F" and "G"), towards a prominent sinuous speed discontinuity extending northward from the peninsula ("H"). This discontinuity indicates convergence of roughly 10 cm s$^{-1}$over a distance of less than 100 m, which is expected to lead to large-scale scale rafting and ridge building. However, there is no evidence of any ridges or ice rubble in the backscatter in Figure 6a. This suggests the event had only just commenced and / or only occurred on a timescale of a few seconds. The absence of any ridge

features in the backscatter amplitude imagery suggests the process may be transient, impacting different sections of ice as it passes by the point of convergence.

The Vilkitsky Strait is known for the formation of ice arches in the springtime through the consolidation of ice with m-scale thickness. Ice arches form when ice passing through a narrow passage experiences flow stoppage as a result of confining pressure and behaves like landfast ice (Hibler et al., 2006). The scenario presented here may be the precursor to the formation

of an ice arch where the drifting ice increasingly gets confined leading to temporarily stationary ice. Although, during December, the ice does not possess the thickness and strength to withstand the building pressure. The result is the buildup of stagnant ice under transient stress conditions as the pressure cyclically builds up and is released through ice failure. The general direction of the ice drift in the Vilkitsky Strait can be determined strictly based on the backscatter image (Figure 6a) by evaluating among other the lead at the southern margin of the strait ("I") and the apex of the partial ice arches, which points

upstream ("J"). However, this example illustrates additional important utilities of this approach, namely not only to evaluate general drift direction and speed, but also to distinguish between very different dynamic regimes which cannot be evaluated strictly from the amplitude image. We have also demonstrated the ability to capture short-lived transient dynamics, which would otherwise be invisible if using InSAR with longer (> 1s) time lags.

## 4 Discussion

We have demonstrated the potential use of S-ATI for derivation of instantaneous sea ice drift. The phase-derived speed has shown to conform well with a ground-radar validation dataset with accuracy within roughly 0.1 m s$^{-1}$ for ice floes in free drift. This validation was limited to rigid ice floes as the young, fragmented ice did not result in a consistent backscatter signature that could be tracked with the ground-radar. Also, the contributions to the Doppler velocity can be large and difficult to correct in areas of young ice where the ice motion is impacted by dm-scale waves. A high accuracy of InSAR-derived motion is

expected based on prior InSAR validation over landfast ice using longer interferometric time lags on the scale of days to weeks (Dammann et al., 2018a). However, even if S-ATI is an accurate tool to assess ice drift, it has significant limitations for possible applications.

Interferometric products only resolve one dimension (look direction) of the 2-dimensional drift; hence the actual drift speed cannot be resolved directly from the interferogram without additional interpretation steps. For surfaces experiencing consistent displacement over time periods of several hours, (e.g. glaciers) 2-dimensional motion vectors can be estimated by data from both ascending and descending passes (Lang, 2003). This is generally not the case for sea ice, but Dammann et al. (2016) demonstrated that by evaluating the coastline, persistent drift patterns, and additional coarser resolution datasets, it is possible to narrow down likely directions of ice motion. In the case of fixed installations near the coast, information related to the general drift pattern may be sufficient to determine the true velocity field. An example is Utqiaġvik, where the near-coastal sea ice predominately drifts in the orientation of the coastline. This leaves two possible directions of motion, northeast and southwest, which can be discriminated from the sign of the phase values. Such reconstruction of 2-dimensional motion will not always be reliable. However, considering wind, currents and general drift information will help decide when conditions for application of S-ATI are close to optimal. In the case of the fracture in the Fram Strait, we extracted the closing direction by comparing with Sentinel-1 imagery (Figure 5). However, even in the absence of other data, a range of likely directions can be estimated based on edge morphology. For instance, shear motion is unlikely along non-linear lead systems.

S-ATI merely provides a snapshot in time, hence the derived drift should be evaluated with caution and preferably used to compliment other SAR-based drift products. For instance, there may be cases where the interferograms capture ice which is being pushed in one direction creating build-up of ice forces leading to short-term rebound effects where ice motion is significantly slowed down or reversed. This may be particularly relevant in areas of high ice concentration where the build-up of internal ice pressure can be substantial. Although such cases will be rare, it is necessary to consider such possibilities to make sure the derived instantaneous drift is representative of the general ice drift. Also, since only one isolated snapshot on the millisecond-scale can be analyzed, short-lived or transient dynamics cannot directly be deducted from S-ATI alone.

It is necessary to consider the impact of topographic contribution to the interferometric phase due to a non-zero perpendicular baseline. In our analysis in the Fram Strait, $h_a$ = 65 m resulting only in a minor phase change from elevation. For instance, if the ice on each side of the fracture would feature a 1 cm freeboard difference, it would only result in a 0.5 mm s$^{-1}$ drift speed error, $v_e$ (Equation 6). However, this image features large multiyear floes (see "A" in Figure 4a and b) causing a phase change of around 0.03 radians from surrounding ice. This implies an elevation change of roughly 30 cm and hence a difference in ice thickness of about 3 m, which is a reasonable difference between multiyear and first-year ice in this region. Icebergs can feature even larger elevation changes than multiyear ice in which cases the sail height can end up dominating the phase signal (Dammann et al., In review) as is the case here leading to values falling outside of the color range and saturating the image (see white areas in "B" in Figure 4b).

The drift speed error from multiyear ice and icebergs can be substantial. For a single resolution cell, we cannot distinguish the influence of height variations from the influence of motion. The inclusion of the amplitude images and the judgement of the neighborhood of each pixel in the analysis of the drift field is necessary to exclude topographic results from the analysis. Small effective baselines will be beneficial due to reduced sensitivity to topography. Hence, the question is how large the acceptable length of the perpendicular baseline, $B_\perp$, is so that the effect of topography (quantified by the maximum height) corresponds to the effect of phase noise on the phase-derived speed. By combining equations (3) and (6) and setting $\sigma_v = v_e$, we can determine $B_\perp$ for a given height $h_0$. Assuming $\theta = 30°$, and $\sigma_\phi$ similar to what is used in our work based on $\gamma = 0.8$, we determine maximum $B_\perp \sim 2$ km for a large $h_0 = 0.3$ m. A small $v_a$ is also beneficial by reducing sensitivity to noise and $v_e$. As $v_a$ is inversely proportional to the along-track baseline, a possible tenfold increase in $B_\parallel$ leading to $B_t \sim 1$ s would likely be more optimal. A further increase in $B_t$ would likely result in increased rotation of floes and possible deformation, further complicating the results.

Even without significant rotation, it may be necessary to consider phase ambiguities in discontinuities. For the example of a closing lead in the Fram Strait, the interferometric phase could be tracked continuously near the southeastern part of the image. However, in a case where a fracture caused a phase discontinuity extending all the way through the image, $\Delta v_\phi$ could theoretically not be determined since multiples of $2\pi$ could not have been discriminated. Even so, considering $v_a$-values reaching upwards of 1 m, would result in implausible $\Delta v_s$-values if the phase discontinuity represented a phase change of more than $2\pi$.

## 5 Conclusion

Sea ice is a significant component of Arctic ecosystems and its dynamic nature is of critical relevance to human near-coastal or offshore activities. Multiple techniques exist to evaluate sea ice drift across large spatial scales using remote sensing, but often with limited accuracy due to the temporal lag between satellite overpasses. We here investigate the potential of single-pass TanDEM-X interferometry (S-ATI) for deriving more accurate instantaneous drift speeds with a m-scale resolution capable of supporting stakeholders. The approach resulted in values roughly within 10 % of validation data in the form of 2.5 h drift speed averages derived from a ground-based radar system in Utqiaġvik, Alaska. The approach was further used to determine the closing speed of a fracture in the Fram Strait. The ability of estimating the separation/closing rate of leads is an application with relevance for transportation since opening of fractures limits over-ice travel, but serves as pathways for ocean navigation.

Lastly, the approach was demonstrated in the Vilkitsky Strait, an important strategic location for trans-Arctic shipping as part of the northern sea route. Here, S-ATI showed capable of discriminating different dynamic regimes and identify zones of shear

and convergence not easily identified in the amplitude image. The case study in the Vilkitsky Strait not only demonstrates the application for InSAR-derived drift speeds, but also the ability to resolve important sea ice processes at a scale and accuracy which have been difficult to assess in the past. As an example, we were able to resolve short-lived transient convergence processes otherwise invisible to SAR approaches. Such detailed information pertaining to drift speed could potentially be used

to accurately determine convergence and divergence in a similar approach as applied to landfast ice. With the m-scale resolution of stripmap X-band SAR, this approach would likely be able to provide statistics of maximum pressure loads on structures relevant for engineering design and planning of offshore installations. Furthermore, instantaneous velocity measurements may provide new insight into how drifting sea ice respond to the surface current and wind fields and how the motion of ice floes differ at a moment in time.

Even with the potential application of S-ATI for evaluating short-lived processes, it will inevitably require a careful analysis of the environmental forcing over a longer time period. Only then it is possible to know whether the phase-derived drift can be representative on scales from minutes to hours. In this context, it would also be beneficial to design experiments to study ice motion on temporal scales from sub-seconds (i.e. S-ATI) to minutes (e.g. coastal radar, buoys) to several hours (e.g. buoys, SAR) to better understand observations at different scales and relate both standard SAR approaches (e.g. maximum cross

correlation) to S-ATI. Furthermore, an important question is what temporal resolution is required to investigate "short-lived" events such as transient convergence or strain response upon ice impact with structures. This may also be detectable on scales of several seconds, but questionable on longer timescales of minutes. It is presently largely an open question what short-lived processes occur at different timescales and what temporal baselines would be best suited to capture them.

The largest limitations of S-ATI are likely related to data availability and the fact that only the cross-track component of drift

speed is captured. The latter results in absolute drift speed being difficult to interpret and potentially invisible if motion is directly in the along-track direction. However, existing spaceborne along-track InSAR systems such as TanDEM-X are predominately used for proof of concept, while future dedicated systems for ocean applications would largely reduce these limitations. For instance, the new satellite concept SEASTAR will be able to provide the 2-dimensional motion vector field. TanDEM-X is presently the only system that can produce consecutive SAR images with ms-scale temporal lag necessary to

derive interferometric estimates of instantaneous sea ice drift speed. However, with potential newer systems such as the proposed TanDEM-L mission, higher temporal resolution of drift estimates may be obtained by utilizing interferograms from multiple sensors.

**Acknowledgements**

This work was supported by the Swedish National Space Agency (Dnr 192/15). TanDEM-X data were provided free of charge

by the German Aerospace Center (DLR) through a science proposal (XTI_GLAC6921). Sentinel-1 data are provided free of charge by the European Union Copernicus program and were accessed through the Alaska Satellite Facility (ASF). The mooring observations by Utqiaġvik was supported by Arctic Research for Sustainability (ArCS) Project and Grant in Aid for

Scientific Research 15H03721 from the Japanese Ministry of Education, Culture, Sports, Science, and Technology. We thank Bill Hauer at ASF, Thomas Busche at DLR, and David Duncan and Anis Elyouncha at Chalmers University of Technology for valuable support and guidance. We thank two reviewers, including Wolfgang Dierking, for substantially improving the manuscript.

**Author contributions**

Dyre Dammann conducted the interferometric processing and analysis and drafted the initial manuscript. Leif Erikson provided critical guidance on all aspects of the analysis and manuscript. Josh Jones and Andrew Mahoney contributed to the collection and interpretation of the ground-based radar data and derived motion products. Roland Romeiser and Franz Meyer provided valuable expertise relevant to interferometric analysis and interpretation. Hajo Eicken provided expertise related to sea ice

deformation and processes in the different study regions. Yasushi Fukamachi contributed to the collection and analysis of the mooring data. All co-authors also provided valuable recommendations and corrections resulting in the final manuscript.

**Competing interests**

The authors declare that they have no conflict of interest.

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

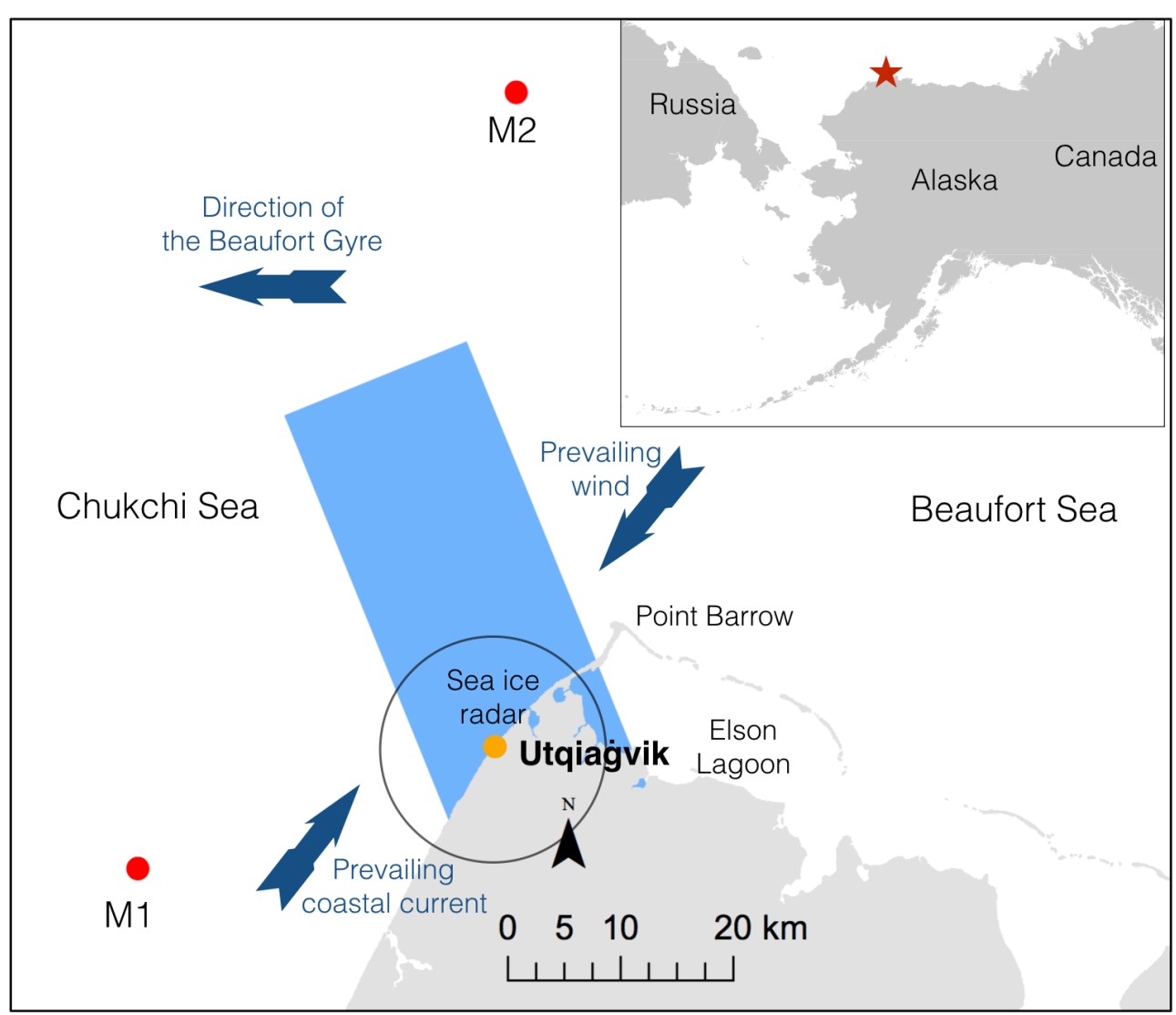

**Figure 1: Overview of study region near Utqiaġvik, Alaska. The blue rectangle signifies the footprint of the SAR acquisitions and the circle marks the range of the ground-based radar (roughly 11 km). Red dots signify location of deployed moorings for assessing ocean surface current.**

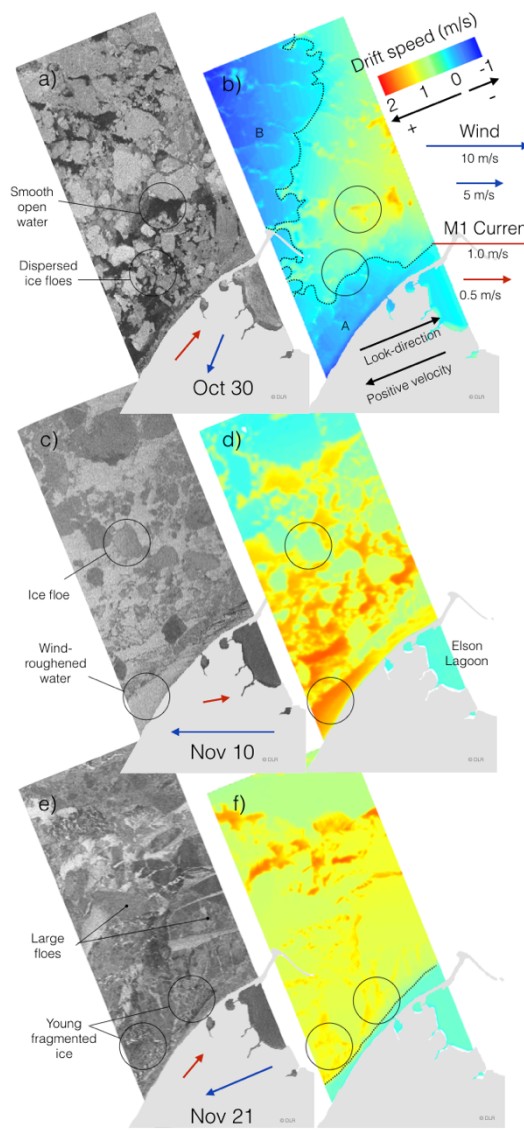

5 **Figure 2: Three TanDEM-X backscatter images individually stretched to emphasize different ice types and features (left column) and phase-derived speed in the look direction (right column) by Utqiaġvik during fall 2015 at 03:19 UTC. Positive velocity is defined opposite of look direction. Line of zero velocity is marked with a dotted line. Velocity of wind (recorded in Utqiaġvik) and currents (at M1) at the time of acquisition are indicated with blue and red arrows respectively. Land is masked out in light gray. A and B indicate areas of negative velocity in (a) further discussed in the text.**

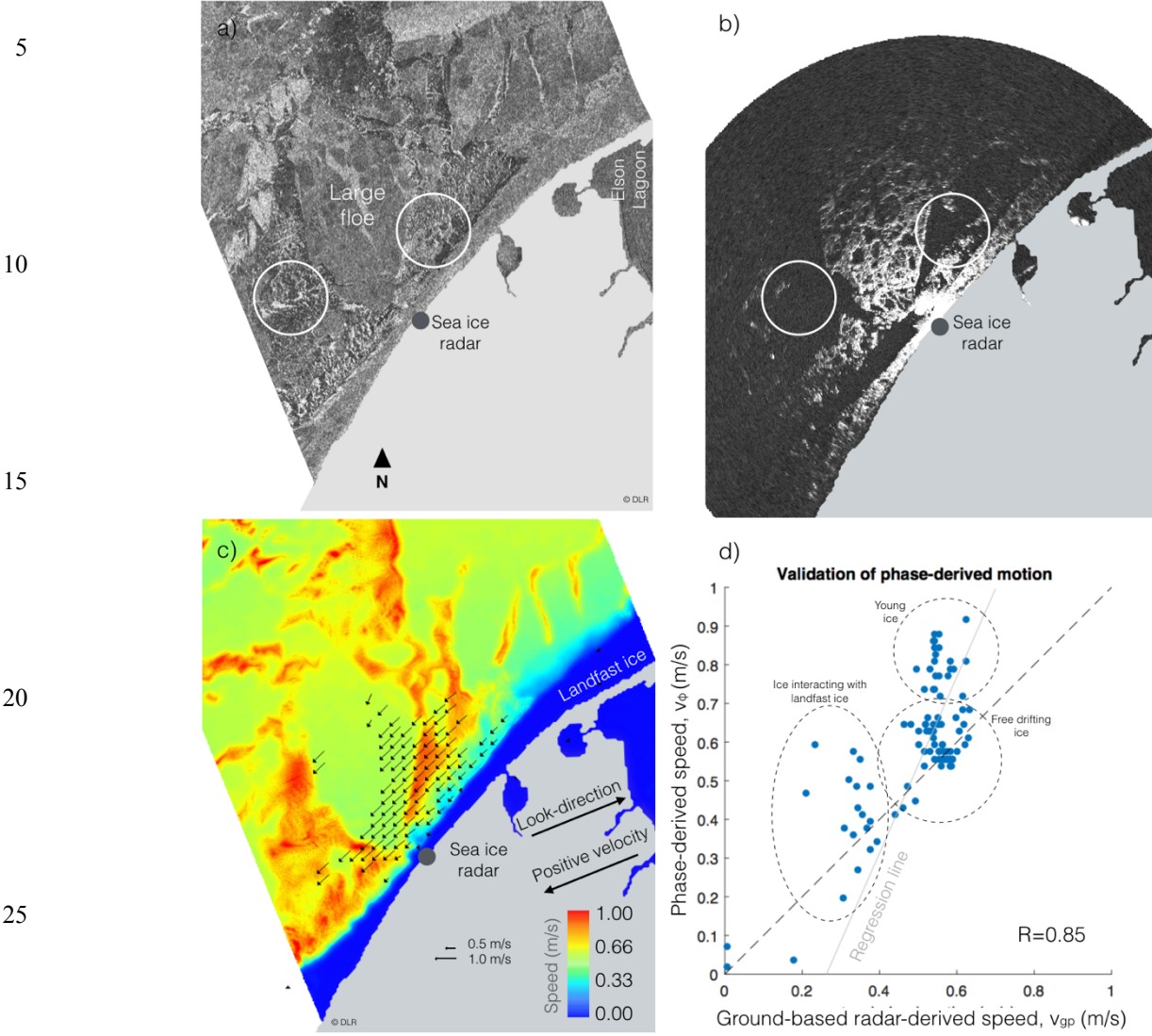

**Figure 3: (a) TanDEM-X backscatter scene over Utqiaġvik, Alaska on 21 Nov 2015 3:19 UTC. (b) ground-based radar backscatter scene 21 Nov 2015 3:18 UTC. Circles indicate areas of young ice hence reduced backscatter in (b). (c) interferometric phase-derived look-directional speed at 3:19 UTC. Arrows represent speed derived from ground-based radar data averaged between 1:52 – 4:28 UTC. Land is masked out in light gray. (d) Comparison between look-directional speed as evaluated using InSAR- and ground-based radar-derived speed. Circles in (d) indicate three types of drifting ice including floes interacting with landfast ice, free drifting floes, and young ice.**

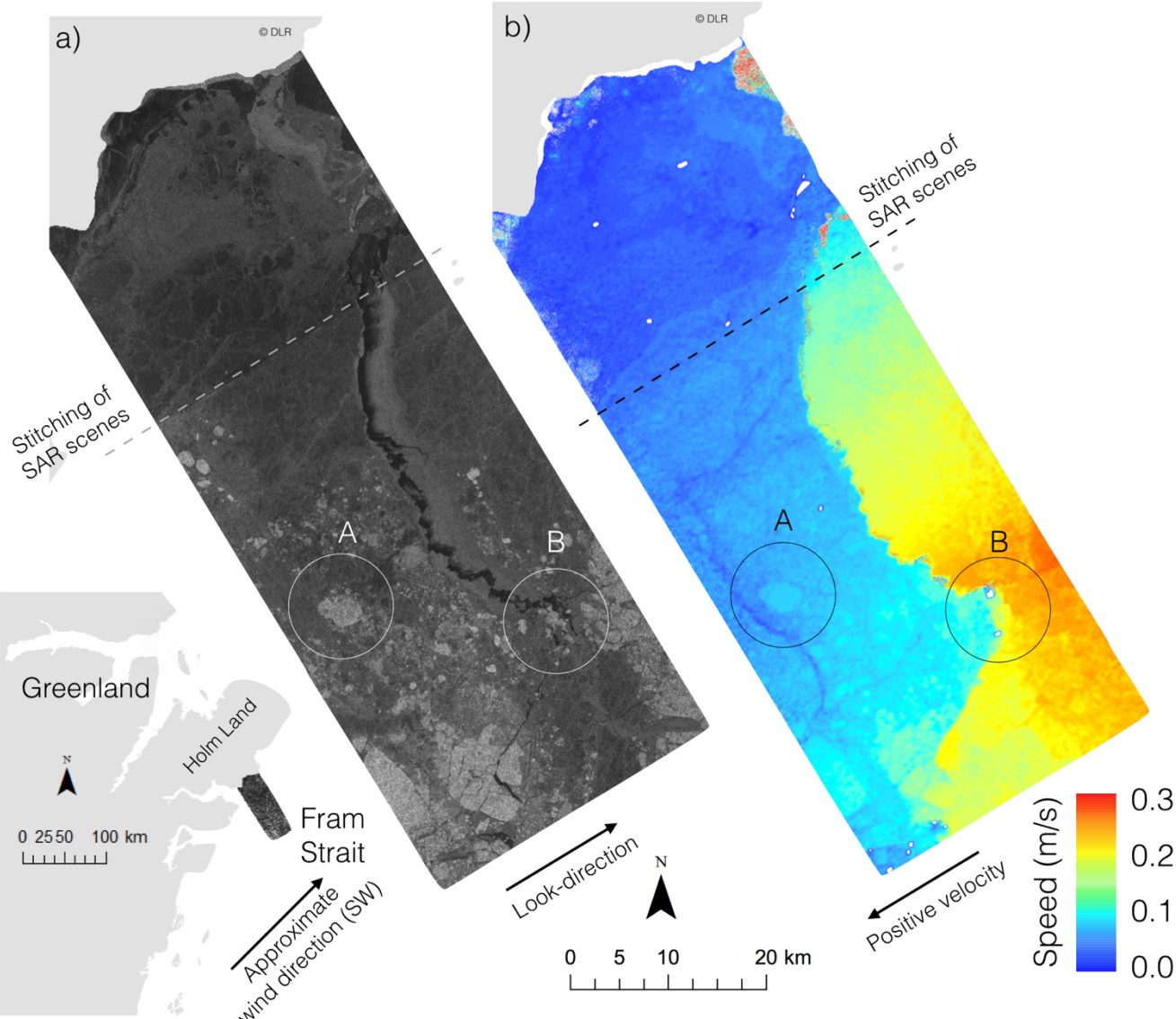

**Figure 4: (a) TanDEM-X backscatter scene over Fram Strait 23 Nov 2015 at 17:00 UTC. (b) Look-directional component of InSAR-derived speed. Letters signify multiyear ice "A" and area of ice bergs "B". Land is masked out in light gray. White areas signify values larger than the range of the color scale due to topography (either icebergs or land topography near the coast due to a poor match with the landmask). Wind at the time of the acquisition was roughly southwesterly at 3 m/s.**

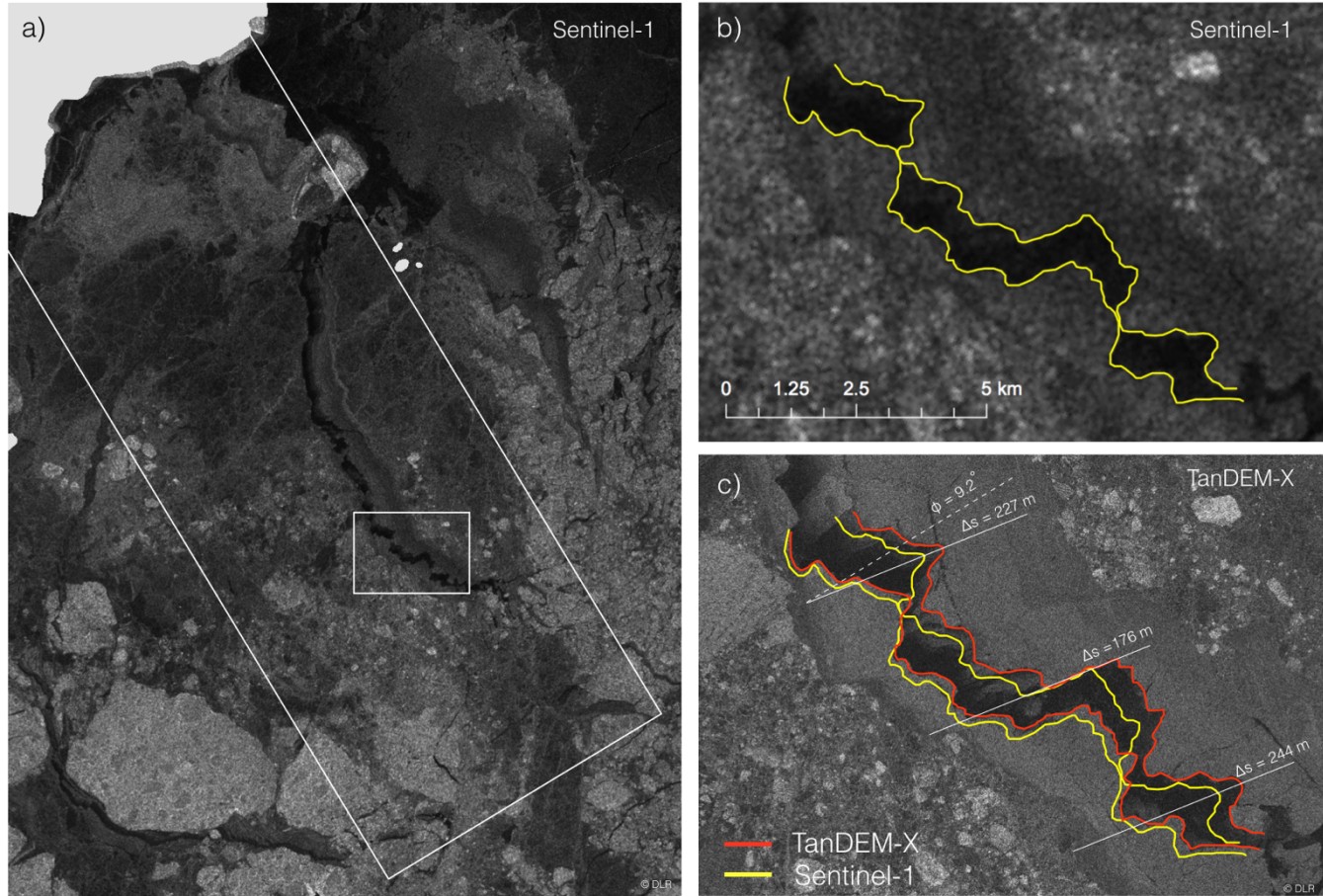

**Figure 5: (a) Sentinel-1 backscatter image acquired 23 Nov 2015 at 17:32 UTC. The large white box represents the areal extent of the TanDEM-X image. (b) outlined part of the fracture (yellow line) as observed with Sentinel-1 within the small rectangle in (a). (c) outlined part of the fracture (red line) as observed with TanDEM-X. Width of the fracture (Δs) is compared along the solid lines and φ represents angle between opening direction (solid lines) and the TanDEM-X look direction (dashed line). Land is masked out in light gray.**

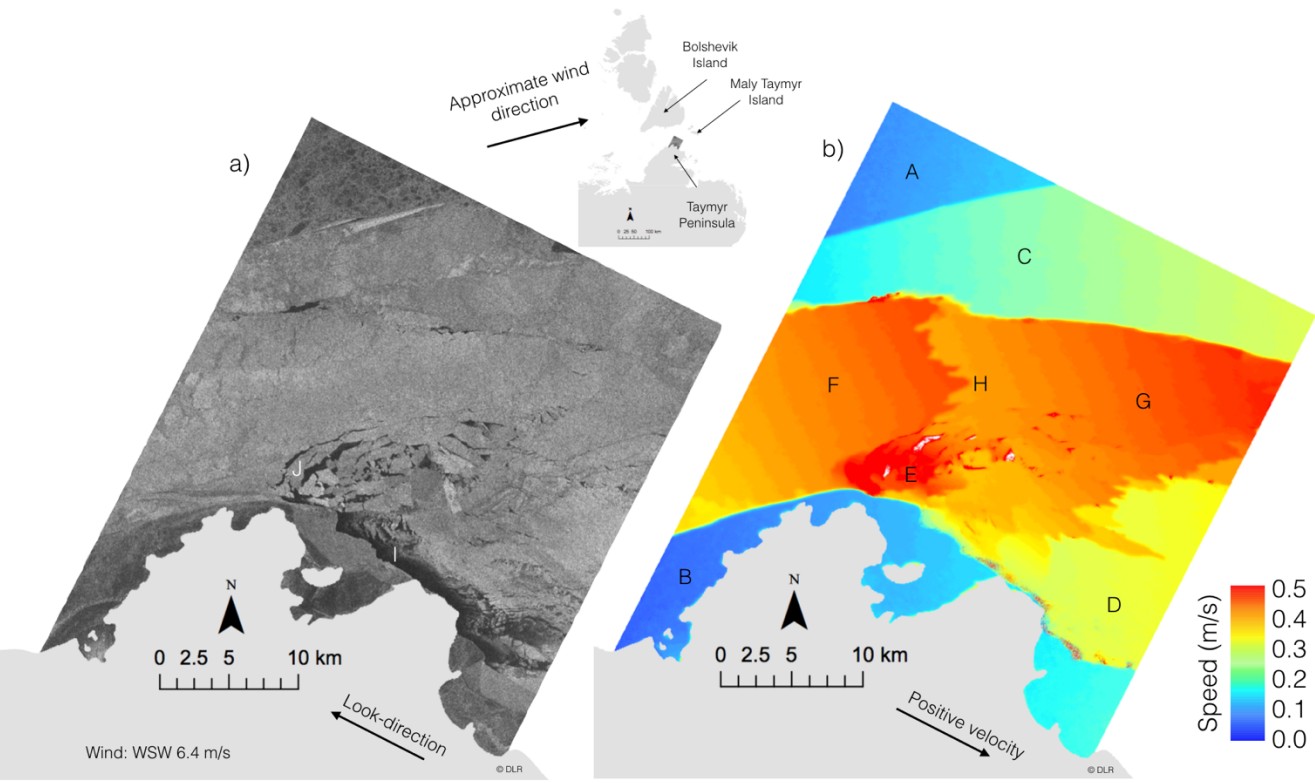

**Figure 6: (a) TanDEM-X backscatter scene over Vilkitsky Strait 17 Dec 2013. (b) Look-directional component of InSAR-derived speed. Land is masked out in light gray. Wind at the time of the acquisition was roughly WSW at 6 m/s. Different zones are indicated by letters (A-B) no drift in the look direction, (C-D) intermediate speeds, and (F-H) channel of high speeds.**

**Table 1: List of TanDEM-X datasets analyzed**

| Region | Acquisition ID | Orbit | Date | Time | Dir. | $B_\parallel$ (m) | $B_\perp$ (m) | $\theta$ (deg) | $v_a$ (m s$^{-1}$) | $h_a$ (m) |
|---|---|---|---|---|---|---|---|---|---|---|
| Utqiaġvik | 1321593 | 46609 | 30 Oct 2015 | 03:19:06.563 | A | 89.2 | 73.8 | 20.9 | 3.70 | 41.6 |
| Utqiaġvik | 1321410 | 46776 | 10 Nov 2015 | 03:19:06.374 | A | 42.6 | 76.4 | 20.9 | 7.76 | 40.2 |
| Utqiaġvik | 1321233 | 46943 | 21 Nov 2015 | 03:19:06.411 | A | 73.3 | 77.4 | 20.9 | 4.51 | 39.7 |
| Fram Strait | 1323154 | 46982 | 23 Nov 2015 | 17:00:48.513 | A | 69.5 | 77.6 | 32.5 | 3.16 | 65.6 |
| Fram Strait | 1323154 | 46982 | 23 Nov 2015 | 17:00:41.513 | A | 69.5 | 78.4 | 32.5 | 3.16 | 64.9 |
| Vilkitsky Strait | 1169973 | 36253 | 17 Dec 2013 | 00:17:00.693 | D | 152.2 | 81.2 | 38.5 | 1.24 | 78.1 |

*Dir.* = orbit direction either ascending (A) or descending (D), Time = acquisition start time in UTC, $B_\parallel$ = along-track baseline, $B_\perp$ = perpendicular baseline, $\theta$ = incident angle, and $v_a$ = speed of ambiguity

