# Peer review of "Instantaneous sea ice drift speed from TanDEM-X interferometry"

_The Cryosphere, 2018_

## Referee Comment (RC1) · Anonymous Referee #1 · 29 Dec 2018

**General comments**

The paper presents nice experiments showing sea ice drift from InSAR using TanDEM-X data:

1. a comparison and validation using a ground based radar in Utquiagvik, Alaska

2. an investigation of fracture dynamics near Holm Land, Greenland

3. drift zones in Vilkitsky Strait, Russia

Though the method is not new, the authors provide a good discussion on interpretation, calibration and errors. They provide realistic arguments for the practical use

of the method along with it's limitations. The authors conclude that InSAR allows resolving meter scale, line-of-sight ice motion which is not resolvable using conventional methods. It is argued that even the relative speed provides insight into transient convergence and divergence of ice, and that this is relevant for ice management and tactical navigation.

Overall, I find the paper to be well structured, well written and of high quality. References are relevant and support the authors' claims. The paper seems like a good fit for this journal and I have only a few comments.

**Specific comments**

- Title:

As noted by the editor, the drift is in the line-of-sight only which is not (explicitly) reflected in the title. The HF-radar is also not mentioned in the title, even though it is a significant part of the paper. I still like the title as it is and leave it to you if you want to take the above aspects into account.

- Page 2, line 22: "The technique further requires incorporation and manual interpretation of land areas in each scene"

Why manual? I would assume that this could be automatically estimated over land. If the absolute antenna behavior is known it should also be possible to correct the measurements even without land. There was a nice paper at EUSAR this year which mentions this: "... This will then allow proper correction also for scenes with insufficient land coverage." Reference: Hansen et al., EUSAR 2018 "A new look at Envisat ASAR range Doppler shift retrieval with the aim of reprocessing ten years of level-0 data".

- Page 3, line 18-19: "to our knowledge has not been used to measure sea ice drift"

I was a little bit surprised by this as your paper already mentions Scheiber2011, who showed some results of TDX InSAR sea ice drift. I think it would be nice to include a reference here and comment on similarities/differences to what they did. As far as I remember they only showed the relative drift without any further calibration/validation, whereas you provide examples of absolute speeds and a more in-depth discussion.

- Page 4, line 1: "... a combination of existing and newly developed methods".

It would be nice to have the essence of the methods provided in the text also. Dense Lucas-Kanade optical flow or something more? Please add a sentence or two to make it more explicit.

**Minor corrections / suggestions**

- Page 3, line 1: "in look direction" -> "in the look direction"

Please correct all occurrences (I might have missed some): Page 5 line 1, Page 5 line 2, Page 8 line 11, Page 8 line 20, Page 9 line 8.

- Page 3, line 29: "...archived every 5 m and ..." -> "5 min." or "5 minutes".

- Page 4, line 21: "... sea ice grown in-situ ..."

The "grown in-situ" part sounds a bit strange to me and could be removed, but perhaps am just not familiar with the use of this phrase. What is it being contrasted with? Grown remotely?

- Page 5, line 3: "...which can be expressed" -> "...which can be expressed as"

- Page 5, line 5: "where theta is the incidence angle" -> "incident angle" or "angle of incidence"

Just a suggestion. Some people are picky about this.

- Page 9, line 20: "formation of ice arches"

The term "ice arch" could perhaps be more precisely specified in the text (I had not heard the term before). I guess it's clear from the context and the fact that your pointing to pont J in the image, but still.

- Page 11, line 11: "features larger multiyear floes" -> "features large multiyear floes"

Larger compared to what?

- Page 16, Figure 1:

It would be nice to have the range in km of the ground-based radar in the caption for easy reference.

- Page 17, Figure 2:

The line of zero velocity is quite weak in the figure which made it disappear completely on my printout. Please try to make it more easily visible, perhaps by making the line thicker or more densely dashed. Was the line drawn manually, or automatically (e.g., from thresholding the velocity map (or similar)?

- Page 18, Figure 3:

I see the right circle in (a) and (b) mentioned in the caption for (d), but what about the left circle? Is it the same type? I could not find the reported R=0.85 value in (d) discussed in the text. Please include a short comment in the text. It would also be nice to have the regression line in the plot along with the existing one-to-one line.

- Page 19, Figure 4:

Please specify in the captions what the white areas in the InSAR speed plot are. I didn't immediately see this commented in the text, sorry if I missed it.

- Page 20, Figure 5:

Consider changing theta to another variable as theta is also used for the angle of incidence, but this is not very important (I guess it's clear enough from context).

- Page 21, Figure 6:

Please include a brief explanation of the regions marked by letters instead of referring to the text. It doesn't have to be long. For example: Different zones are indicated by letters (A-B) no drift in the look direction, (C-D) intermediate speeds, (F-H) channel of high speeds. As a reader I find it annoying to jump back and forth alot between the text and the figure.

- Page 22, Table 1:

There is a lot of white space in the table. Peraps it could be made a bit neater (e.g., consistent spacing around "/", expand Str. into Strait, add [deg] after theta), but this is a very minor issue. Is date/orbit/cycle enough to uniquely identify the frames you used for the experiment or would a timestamp or frame identifier be needed?

---

## Author Comment (AC1) · 11 Jan 2019

General comments

The paper presents nice experiments showing sea ice drift from InSAR using TanDEMX
data:
1. a comparison and validation using a ground based radar in Utquiagvik, Alaska
2. an investigation of fracture dynamics near Holm Land, Greenland
3. drift zones in Vilkitsky Strait, Russia

Though the method is not new, the authors provide a good discussion on interpretation,
calibration and errors. They provide realistic arguments for the practical use of the method along
with it's limitations. The authors conclude that InSAR allows resolving
meter scale, line-of-sight ice motion which is not resolvable using conventional
methods. It is argued that even the relative speed provides insight into transient convergence
and divergence of ice, and that this is relevant for ice management and tactical
navigation.

Overall, I find the paper to be well structured, well written and of high quality. References
are relevant and support the authors' claims. The paper seems like a good fit
for this journal and I have only a few comments.

Specific comments
- Title:
As noted by the editor, the drift is in the line-of-sight only which is not (explicitly) reflected
in the title. The HF-radar is also not mentioned in the title, even though it is a
significant part of the paper. I still like the title as it is and leave it to you if you want to
take the above aspects into account.

Thank you for providing these points. We might lean towards the option of keeping it as is to keep it short and concise although we agree it could be changed to make it more specific.

- Page 2, line 22: "The technique further requires incorporation and manual interpretation of land areas in each scene"
Why manual? I would assume that this could be automatically estimated over land.

Good point. Taken out.

If the absolute antenna behavior is known it should also be possible to correct the measurements even without land. There was a nice paper at EUSAR this year which mentions this: "... This will then allow proper correction also for scenes with insufficient land coverage." Reference: Hansen et al., EUSAR 2018 "A new look at Envisat ASAR range Doppler shift retrieval with the aim of reprocessing ten years of level-0 data".

Thank you for providing this reference. This has now been included (P2,L22): "The technique has in the past required incorporation and interpretation of land areas in each scene where biases can be corrected (Kræmer et al., 2015), but may not be necessary in the future (Hansen et al., 2018). "

- Page 3, line 18-19: "to our knowledge has not been used to measure sea ice drift"
I was a little bit surprised by this as your paper already mentions Scheiber2011, who showed some results of TDX InSAR sea ice drift. I think it would be nice to include a reference here and comment on similarities/differences to what they did. As far as I remember they only showed the relative drift without any further calibration/validation, whereas you provide examples of absolute speeds and a more in-depth discussion.

This has now been better clarified by some small changes to the text (P3,L13): "The fringes resulting from 1-dimensional phase information can be interpreted into 2-dimensional motion using inverse modeling and provide important information pertaining to internal deformation of the ice (Dammann et al., 2016). However, the fringes can at the same time hide a constant phase value pertaining to the general drift speed, which was the case for Scheiber et al. (2011). In contrast, we here apply bistatic acquisitions with substantially shorter (~ 10 ms) temporal lag for the evaluation of instantaneous sea ice drift speed."

- Page 4, line 1: "... a combination of existing and newly developed methods".
It would be nice to have the essence of the methods provided in the text also. Dense Lucas-Kanade optical flow or something more? Please add a sentence or two to make it more explicit.

This has been included. Changed to (P4,L2): "These include dense and feature-based optical flow approaches to compute motion fields from the images, active contours for delineation of stable landfast ice, and Hidden Markov Models for machine learning based event detection (Rohith et al., 2013;Jones et al., 2016). The filtering approach results in motion products averaged over 1.5 hours. For the three cases evaluated here the motion tracking algorithm only produced consistent motion vectors suitable for validation in one case (Nov 21)."

Minor corrections / suggestions

- Page 3, line 1: "in look direction" -> "in the look direction"
Please correct all occurrences (I might have missed some): Page 5 line 1, Page 5 line 2, Page 8 line 11, Page 8 line 20, Page 9 line 8.

Done

- Page 3, line 29: "...archived every 5 m and ..." -> "5 min." or "5 minutes".

Done

- Page 4, line 21: "... sea ice grown in-situ ..."
The "grown in-situ" part sounds a bit strange to me and could be removed, but perhaps am just not familiar with the use of this phrase. What is it being contrasted with? Grown remotely?

Taken out in-situ as we explain the potential for thicker ice advected from other places.

- Page 5, line 3: "...which can be expressed" -> "...which can be expressed as"

Done

- Page 5, line 5: "where theta is the incidence angle" -> "incident angle" or "angle of incidence"
Just a suggestion. Some people are picky about this.

Done

- Page 9, line 20: "formation of ice arches"
The term "ice arch" could perhaps be more precisely specified in the text (I had not heard the term before). I guess it's clear from the context and the fact that your pointing to pont J in the image, but still.

Good point. Included (P9,L26): "Ice arches form when ice passing through a narrow passage experiences flow stoppage as a result of confining pressure and behaves like landfast ice (Hibler et al., 2006)."

- Page 11, line 11: "features larger multiyear floes" -> "features large multiyear floes"
Larger compared to what?

Done

- Page 16, Figure 1:
It would be nice to have the range in km of the ground-based radar in the caption for easy reference.

Included

- Page 17, Figure 2:
The line of zero velocity is quite weak in the figure which made it disappear completely
on my printout. Please try to make it more easily visible, perhaps by making the line
thicker or more densely dashed. Was the line drawn manually, or automatically (e.g.,
from thresholding the velocity map (or similar)?

Done. This was drawn automatically as the zero contour

- Page 18, Figure 3:
I see the right circle in (a) and (b) mentioned in the caption for (d), but what about
the left circle? Is it the same type?
Yes, now changed to: "circled areas"

I could not find the reported R=0.85 value in (d) discussed in the text. Please include a short
comment in the text.

Included (P7,L29): "The v_gr confirms largely homogenous drift speed of the floes in the
southwest direction corresponding to v_ϕ (Figure 3d) with an R-value of 0.85."

It would also be nice to have the regression line in the plot along with the existing one-to-one
line.

Done

- Page 19, Figure 4:
Please specify in the captions what the white areas in the InSAR speed plot are. I
didn't immediately see this commented in the text, sorry if I missed it.

Now included: "White areas signify values larger than the range of the color scale due to
topography (either icebergs or land topography near the coast due to a poor match with the
landmask)."

- Page 20, Figure 5:
Consider changing theta to another variable as theta is also used for the angle of
incidence, but this is not very important (I guess it's clear enough from context).

Good catch. Changed to φ

- Page 21, Figure 6:
Please include a brief explanation of the regions marked by letters instead of referring
to the text. It doesn't have to be long. For example: Different zones are indicated by
letters (A-B) no drift in the look direction, (C-D) intermediate speeds, (F-H) channel of
high speeds. As a reader I find it annoying to jump back and forth alot between the text
and the figure.

Good idea. Done

- Page 22, Table 1:
There is a lot of white space in the table. Peraps it could be made a bit neater (e.g., consistent spacing around "/", expand Str. into Strait, add [deg] after theta), but this is a very minor issue. Is date/orbit/cycle enough to uniquely identify the frames you used for the experiment or would a timestamp or frame identifier be needed?

We have cleaned this up. We have now included the acquisition start time and ID.

[revised manuscript text omitted]

---

## Referee Comment (RC2) · Dierking (Referee) · 16 Jan 2019

**Review of „Instantaneous sea ice drift speed from TanDEM-X interferometry"**
**by D. O. Dammann and 7 co-authors (original manuscript)**

**General assessment:**
From a technological point of view, the paper is very interesting. The shown case studies represent different scenarios for which the results of single-pass along-track interferometry (S-ATI) can potentially contribute to extend our understanding of sea ice dynamics for specific conditions. However, I disagree strongly with the author's assessment concerning the benefit of instantaneous drift fields for different applications. The technique provides useful results only if the ice drift component (anti)parallel to the line-of-sight between radar and resolution cell is considerably larger than the drift component perpendicular to it. If the line-of-sight drift component is zero, along-track InSAR is blind to sea ice movements. Besides this limitation of information on direction it is not clear how representative a single snapshot of millisecond ice movement is for the temporal variations of an ice drift field over several minutes to hours.
Since I regard the topic of the paper and the scientific analysis (section 3) as important contribution to the use of InSAR for sea ice research, I recommend a major revision of the paper to sharpen its message and to provide the reader with a clear list of pros and cons of the technology and still existing gaps of knowledge.

**Details**

**Abstract:**
In the abstract, not a single word is related to the inherent limitations of the method. Why is the knowledge of the millisecond ice drift line-of-sight (LOS) velocity component of advantage for ice management and transportation if compared to standard SAR-based drift algorithms? I doubt this since (1) the field of instantaneous drift components for a given location can be obtained only with time gaps of hours to days between single data takes, (2) the standard SAR-based drift algorithms have the potential to retrieve 2D displacement information with higher temporal resolution if the time gaps between (non-tandem) SAR data acquisitions get shorter. This will be the case e.g. in satellite constellations, or by using images of different SAR systems - a combination of C- and L-band was already successfully tested. I could imagine that for engineering, instantaneous drift can be a valuable information to get statistics of extreme values, but sea ice modellers prefer to get magnitude and direction of drift vectors, and time steps used in simulations of sea ice dynamics are not on the order of milliseconds.

**Introduction**
p. 2, 15-23: It is of course true that complex drift patterns often cannot be resolved into sufficiently small time steps by traditional ice drift retrievals. However, using the Doppler-approach described by Kræmer et al. (2015, 2018) or S-ATI as described here is NOT the solution to this problem, since, as described above, one obtains milliseconds snapshots of ice movement only with large time gaps between single shots. One motivation for Kræmer et al. was to make use of a single SAR image for obtaining information on ice drift – with the same limitations as for S-ATI plus a relatively coarse spatial resolution. In lines 20-22 the authors list factors that influence the accuracy of the Doppler method, but in the following sections I did not find a similar list for the S-ATI method (see below). The Doppler method requires the identification of land areas, but so does S-ATI (which the authors describe on p. 5, 9-10 – here they use landfast ice as reference).
p. 2, 24-29: why is the method described here "consistent" but the other methods are not? Please explain what you mean by "consistent". Which requirements do Arctic stakeholders have with regards to spatial resolution of drift fields? Any reference? On regional scales of a few hundred kilometres, a spatial resolution of drift vector fields around 100 m (which is possible to achieve with traditional methods) is certainly sufficient for many applications. The only ice services, which started to consider results of automated drift retrievals in ice chart production, is to my knowledge CIS. Why and for what specific purpose do shipping and resource exploration need drift fields with a meter-resolution? How is drifting pack ice "locally used"? (For traffic on landfast ice, information on differential motion with resolutions of meters is of course beneficial). Is S-ATI really the full

answer when it comes to gather "high-resolution" data of ice hazard distributions, ice movement etc.? It must be taken into account that it often fails to deliver useful data, see above.

The last paragraph of the introduction should be moved to section 2.3, since it specifically addresses issues of InSAR processing.

p. 3 11-13 If the temporal baseline is too large (the limit dependent on ice speed and environmental conditions changing scattering properties), the interferometric coherence over sea ice is lost. Dependent on drift speed and SAR resolution, this can be minutes.

p.3 13-15 Inverse modelling to provide 2D motion from the line-of-sight drift can be used for landfast ice but not for drifting sea ice.

**Section 2**

p. 3 29: Radar images are archived every *5 m*…??

What are the uncertainties in velocity as determined from coastal radar and ADCP data?

p. 4, 1-3: Regarding the coastal radar - the motion product represents an average field over a time interval of 1.5 hours. How large is the noise contribution? Would it be possible to calculate motion for much smaller time intervals to judge short(er)-term variations of the drift field that could help to estimate a time interval for which millisecond ice movements derived from S-ATI can be representative?

Table 1: it would be useful to add the perpendicular baselines (for use in equation 3) and to mention the spatial resolution of the filtered interferogram used for calculating the LOS drift speed in the text.

p. 4: at the end of the sentence following equation 1, another hint to Table 1 ($v_a$) would be useful.

p. 4, after line 14: The expected magnitudes of phase noise and the resulting relative error of the retrieved values of $v_\phi$ should be discussed. See also "p. 8, 11" below.

p. 4, equation 4: Values of $h_0$ should also be listed in Table 1.

p. 4, 25-26 and p. 5, 1-2: important point! Here, the information on the spatial resolution of the derived drift field would again be useful. For a single resolution cell, we cannot distinguish the influence of height variations from the influence of motion. The inclusion of the amplitude images and the judgement of the neighbourhood of each pixel in the analysis of the drift field is required as the authors demonstrate in their case studies. This should be made clear, e. g. in the discussion.

p. 5: last paragraph of 2.4: good point! What about rotational motion of single ice floes that might have occurred in mixtures of smaller ice floes in open water that appear in your examples?

**Section 3:**

Here one should note that (naturally) optimal conditions were selected as examples: Except for Oct 30 it can be expected that the respective dominant ice speed component is the one parallel to the LOS.

p. 7 12-13 "…is consistent with…." I assume that the authors did not directly observe the amplitude increase of dm-scale wind waves with distance from the shore but merely speculate that this could be the case. Do you have a reference concerning observations of the increase of amplitudes with distance from the coast? Is it also possible that streaks of frazil and grease ice were present? With the given wind direction in Fig. 2c, it would have accumulated at the shore, thus reducing the retrieved LOS drift velocity.

p.7 19: Here one should start a new paragraph: " We further compare…"

p. 7 23-24: It is misleading that in the first sentence $v_{gr}$ is used for the drift vector derived from the coastal radar, and in the second sentence for the LOS-component "$v_{gr}$" derived from $v_{gr}$.

p. 7 26: Here the averaging time for the coastal radar is given as 2.5 hours, in section 2.1, 1.5 hours were mentioned.

p.8, 1-2: the drift speed error mentioned here is only related to the motion-height ambiguity. But phase noise also contributes to the drift speed error.

p. 8, 2-4: Why is a difference between landfast and adjacent drifting ice of 5 cm not possible? I interpret the SAR amplitude image in Fig. 3 such that there is a mixture of smaller dark appearing (hence thinner) ice floes in open water (bright) - in the middle interrupted by more consolidated thin

ice - next to the landfast ice. However, since the image is relatively small in the manuscript, it is difficult to interpret it.

p. 8, 11: the phase cannot be solved accurately, it includes an uncertainty due to phase noise (which depends on decorrelation effects) and the achievable accuracies in determination of baselines, incidence angles, co-registration in processing etc. These factors are not mentioned in section 2 or in the discussion.

p. 8, 15-17: "….sea ice  and features a…" Isn't it "which features"?

p. 9, 16-19: This interpretation again raises the question about how representative milliseconds snapshots are on time-scales of minutes to hours. Convergence should be given in velocity per length unit. The identification of ridges and rubble fields is relatively difficult at higher frequencies such as X-band, unless the spatial resolution is high (another reason to mention it in the paper).

p. 9, 29-30: Since only one temporally isolated snapshot of milliseconds LOS velocity is available, *short-lived (or transient)* dynamics cannot directly be deduced from the data alone. This requires a careful additional analysis of the environmental context over a longer time period. Without such an analysis one does not know whether an event lasts for seconds, minutes or hours. It also needs to be clarified what "short-lived" means in terms of time units. Certainly not milliseconds.

**Section 4**

p. 10, 1-2: "high accuracy" – the plot Fig. 3 d does not support this statement, since 2.5 hours average velocities are compared with milliseconds ice movements. It looks like a "reasonable" agreement between both velocities, but with maximum deviations up to 0.3-0.4 m/s even when excluding the young ice values. It would be helpful to get examples of the relative error for different velocity ranges.

p. 10, first paragraph: the limitations of S-ATI have to be accounted for in this discussion!

- Examples: Opening and closing rates for leads can only be provided for certain orientations between lead and LOS, and information on those rates is only available for the past – for navigation such information may be useful - but as forecast.
- Is a millisecond snapshot well suited to separate landfast and temporally stationary ice – and would this really improve ice charting? The duration to classify stationary ice as landfast is often fixed at 20 days. What is then "temporally stationary"? How can a practical approach to distinguish landfast and temporally stationary ice by means of S-ATI look like?

p. 10, second paragraph: a reconstruction of the 2D instantaneous drift field from the 1D-LOS component will not be very reliable but the consideration of wind, current and general ice drift information will help to decide when conditions for the application of S-ATI are close to optimal.

p. 11, first paragraph: I regard this example of deriving absolute speed from a moving vessel and relative speed between vessel and surrounding ice as an interesting scientific exercise but not useful for ice navigation. The example of protecting offshore installations should be elaborated but I presume that it is more a question of collecting statistics of maximum pressure loads on planned installations instead of giving sporadic information to already existing installations.

p.11, third paragraph: No, the S-ATI approach cannot be more accurate than the traditional approaches because of the reasons mentioned above. They are two very different, complementary approaches. It would be worth to design experiments to study ice motion on temporal scales from sub-seconds (S-ATI) to minutes (coastal radar, buoys) to several hours (buoys, SAR constellations) to better relate both approaches to one another.

**Conclusions:**

Comments given above should be considered.

p. 12, 19-20: Given (1) the (yet) missing link between millisecond and hours timescales, (2) the difficult interpretation related to the missing information on drift direction, and (3) the lack of tandem-missions useful for operational ice charting it is NOT clear that S-ATI is relevant as a tactical tool for Arctic stakeholders.

**Recommendations:**

It is not up to the reviewer to demand changes of the paper contents. Nevertheless, I would like to recommend some items for consideration in the discussion:

- It would be useful to compare the approach of Kræmer et al. and the S-ATI technique in some detail since they have similar limitations concerning certain aspects but are also different in other aspects. Move lines 19-23 on p. 2 into the discussion.
- In the Dierking et al. paper the goal was to retrieve sea ice topography, and the influence of ice motion on the interferometric phase was a disturbing factor that needs to be minimized in this type of application. Here, it is vice versa: motion as wanted information and topography as disturbing factor. Which along- and across-track baselines would you recommend, considering the typical ranges of ice drift speed and height elevations?
- What is the highest temporal resolution that is required for investigating "short-lived" events? Examples of such events? I am sure that we can extrapolate the milliseconds measurements into the range of seconds, but what are the highest possible frequencies of motion changes, and to which events are they linked? This item is very tricky to answer, and it may be sufficient at this time just to raise these questions to make the reader aware of still existing problems.
- Since the interpretation of drift fields derived from S-ATI is not straightforward, it may be useful for the reader to get a "recipe" of important factors to be considered (e .g. uncertainty of the measurement which depends on phase noise and along-track baseline, LOS – where to get information of the main drift direction, millisecond snapshot – what has to be expected for the next minutes/hours etc).

---

## Author Comment (AC2) · 9 Feb 2019

Review of „Instantaneous sea ice drift speed from TanDEM-X interferometry"
by D. O. Dammann and 7 co-authors (original manuscript)

General assessment:
From a technological point of view, the paper is very interesting. The shown case studies represent different scenarios for which the results of single-pass along-track interferometry (S-ATI) can potentially contribute to extend our understanding of sea ice dynamics for specific conditions. However, I disagree strongly with the author's assessment concerning the benefit of instantaneous drift fields for different applications. The technique provides useful results only if the ice drift component (anti)parallel to the line-of-sight between radar and resolution cell is considerably larger than the drift component perpendicular to it. If the line-of-sight drift component is zero, along-track InSAR is blind to sea ice movements. Besides this limitation of information on direction it is not clear how representative a single snapshot of millisecond ice movement is for the temporal variations of an ice drift field over several minutes to hours.
Since I regard the topic of the paper and the scientific analysis (section 3) as important contribution to the use of InSAR for sea ice research, I recommend a major revision of the paper to sharpen its message and to provide the reader with a clear list of pros and cons of the technology and still existing gaps of knowledge.

Dear Professor Dierking,

Thank you so much for providing a thorough review and valuable suggestions for this manuscript. We fully agree with your assessment and have made changes according to your suggestions. The changes have resulted in omitting claims that S-ATI is valuable for tactical decision making, while at the same time introducing other potential uses. We have also made it clearer that this is a proof-of-concept with better application with future new systems. The changes have resulted in substantial changes to the discussion and conclusion sections and overall a new and improved manuscript.

Thank you again!

Best regards,
Dyre Dammann

Details
Abstract:
In the abstract, not a single word is related to the inherent limitations of the method. Why is the knowledge of the millisecond ice drift line-of-sight (LOS) velocity component of advantage for ice management and transportation if compared to standard SAR-based drift algorithms? I doubt this since (1) the field of instantaneous drift components for a given location can be obtained only with time gaps of hours to days between single data takes, (2) the standard SAR-based drift algorithms have the potential to retrieve 2D displacement information with higher temporal resolution if the time gaps between (non-tandem) SAR data acquisitions get shorter. This will be the case e.g. in satellite constellations, or by using images of different SAR systems - a combination of C- and Lband was already successfully tested. I could imagine that for engineering, instantaneous drift can be a valuable information to get statistics of extreme values, but sea ice modellers prefer to get magnitude and direction of drift vectors, and time steps used in simulations of sea ice dynamics are not on the order of milliseconds.
Introduction

Thank you for providing these comments. We very much agree. We have now changed the abstract and addressed all your concerns.

p. 2, 15-23: It is of course true that complex drift patterns often cannot be resolved into sufficiently small time steps by traditional ice drift retrievals. However, using the Doppler-approach described by Kræmer et al. (2015, 2018) or S-ATI as described here is NOT the solution to this problem, since, as described above, one obtains milliseconds snapshots of ice movement only with large time gaps between single shots. One motivation for Kræmer et al. was to make use of a single SAR image for obtaining information on ice drift – with the same limitations as for S-ATI plus a relatively coarse spatial resolution.

Agree. This value of InSAR or DCA as stand-alone product is toned down and we now state (P2,L19): "Other SAR-based approaches, such as Doppler centroid anomaly (DCA) can provide instantaneous ice drift speed, but with other inherent limitations (Kræmer et al., 2015). Instantaneous drift estimates can possibly be used to supplement traditional SAR-based ice drift algorithms for improved accuracy."

In lines 20-22 the authors list factors that influence the accuracy of the Doppler method, but in the following sections I did not find a similar list for the SATI method (see below). The Doppler method requires the identification of land areas, but so does S-ATI (which the authors describe on p. 5, 9-10 – here they use landfast ice as reference).

Good point. Instead of elaborating on DCA, we have removed the list of specific limitations. Instead, we mention S-ATI as a similar approach and further elaborate its limitation in the discussion.

p. 2, 24-29: why is the method described here "consistent" but the other methods are not? Please explain what you mean by "consistent".

Taken out

Which requirements do Arctic stakeholders have with regards to spatial resolution of drift fields? Any reference? On regional scales of a few hundred kilometres, a spatial resolution of drift vector fields around 100 m (which is possible to achieve witht raditional methods) is certainly sufficient for many applications. The only ice services, which started to consider results of automated drift retrievals in ice chart production, is to my knowledge CIS. Why and for what specific purpose do shipping and resource exploration need drift fields with a meter-resolution? How is drifting pack ice "locally used"? (For traffic on landfast ice, information on differential motion with resolutions of meters is of course beneficial).

Good point. We agree that m-scale resolution is not critical for most applications. We have taken the mention of this out and focus now largely on the possibility of assessing short-lived processes. As an example, we have mentioned the assessment of ice response to interaction with structures, relevant for engineering design.

Is S-ATI really the full answer when it comes to gather "high-resolution" data of ice hazard distributions, ice movement etc.? It must be taken into account that it often fails to deliver useful data, see above.

No, we agree we overstated this. This is now significantly toned down and limitations are discussed in greater detail in the discussion.

The last paragraph of the introduction should be moved to section 2.3, since it specifically addresses issues of InSAR processing.

Good point. This has now been merged with Section 2.3.

 If the temporal baseline is too large (the limit dependent on ice speed and environmental conditions changing scattering properties), the interferometric coherence over sea ice is lost. Dependent on drift speed and SAR resolution, this can be minutes.

Included (P4,L26): "With temporal baselines on the order of 10 s (i.e. TanDEM-X pursuit mode), floes can rotate or slightly deform leading to non-homogenous phase values and possibly loss of coherence."

p.3 13-15 Inverse modelling to provide 2D motion from the line-of-sight drift can be used for landfast ice but not for drifting sea ice.

Good point. We agree. Taken out.

Section 2

p. 3 29: Radar images are archived every 5 m…??

Changed to "minutes"

What are the uncertainties in velocity as determined from coastal radar and ADCP data?

We have now included (P3,L19): "For features trackable over much of the radar image, errors in the dislocation vectors are on average well below 10% (Rohith et al., 2013) where an average 5 % error is attributed to uncertainties related to spatial scale and time intervals (Mahoney et al., 2015b)."

And (P3,L31):

"The moorings contained a Teledyne RDI Workhorse Sentinel acoustic Doppler current profiler (ADCP), whose data we use to evaluate surface current velocity (Mahoney et al., 2015a) with an accuracy of less than 0.01 m s-1 (Fukamachi et al., 2006)."

p. 4, 1-3: Regarding the coastal radar - the motion product represents an average field over a time interval of 1.5 hours. How large is the noise contribution?

It is a few percent. Rohith et al., 2013 states: "In summary, for features trackable across much of the image, the corresponding error in the dislocation vector is on average well below 10% of the tracked length. Since these are zero-mean errors, velocity fields obtained for daily or longer intervals are much smaller and associated with errors of a few percent at most."

Would it be possible to calculate motion for much smaller time intervals to judge short(er)-term variations of the drift field that could help to estimate a time interval for which millisecond ice movements derived from S-ATI can be representative?

We attempted a shorter window. We have now mentioned (P3,L28): "We also attempted validation using a 15-minute interval, which is the shortest possible interval using three images, but this resulted in a noisy result."

However, even if steps could be taken to further filter the noisy result, the shortest possible interval is still 15 minutes, which is possibly not much more representative for the millisecond ice movement since we

identified quite consistent flow over the entire day. The good match with the 2.5 h correlation also shows this.

Table 1: it would be useful to add the perpendicular baselines (for use in equation 3) and to mention the spatial resolution of the filtered interferogram used for calculating the LOS drift speed in the text.

Done

p. 4: at the end of the sentence following equation 1, another hint to Table 1 (va) would be useful.

Done

p. 4, after line 14: The expected magnitudes of phase noise and the resulting relative error of the retrieved values of $v\phi$ should be discussed. See also "p. 8, 11" below.

A discussion about the phase noise and resulting velocity error has been included.

p. 4, equation 4: Values of h0 should also be listed in Table 1.

Included

p. 4, 25-26 and p. 5, 1-2: important point! Here, the information on the spatial resolution of the derived drift field would again be useful. For a single resolution cell, we cannot distinguish the influence of height variations from the influence of motion. The inclusion of the amplitude images and the judgement of the neighbourhood of each pixel in the analysis of the drift field is required as the authors demonstrate in their case studies. This should be made clear, e. g. in the discussion.

Grid spacing of ground radar has now been included in Section 2.1. We have also expanded upon the discussion as you suggest (P11,L29): "The drift speed error from multiyear ice and icebergs can be substantial. For a single resolution cell, we cannot distinguish the influence of height variations from the influence of motion. The inclusion of the amplitude images and the judgement of the neighborhood of each pixel in the analysis of the drift field is necessary to exclude topographic results from the analysis."

p. 5: last paragraph of 2.4: good point! What about rotational motion of single ice floes that might have occurred in mixtures of smaller ice floes in open water that appear in your examples?

This has now been described in the beginning of Section 2.4 (P6,L2): "So far, we have strictly considered phase values attributed to linear drift leading to homogenous phase values for individual floes. Additional rotational motion may need to be considered in certain cases. This will result in an along-track phase gradient across rotating floes. If this is observed, the average phase value of the floe can be used to describe the linear motion."

Section 3:
Here one should note that (naturally) optimal conditions were selected as examples: Except for Oct 30 it can be expected that the respective dominant ice speed component is the one parallel to the LOS.

This has been included (P7.L4): "These acquisitions were selected based on optimal conditions, incorporating multiple drifting ice floes and narrow landfast ice extent."

p. 7 12-13 "…is consistent with…." I assume that the authors did not directly observe the amplitude increase of dm-scale wind waves with distance from the shore but merely speculate that this could be the case. Do you have a reference concerning observations of the increase of amplitudes with distance from the coast? Is it also possible that streaks of frazil and grease ice were present? With the given wind direction in Fig. 2c, it would have accumulated at the shore, thus reducing the retrieved LOS drift velocity.

This has now been included (P7,L29): "The apparent speed increase with distance from shore within the bottom circled area in Figure 2d, which would be consistent with plausible dm-scale wind-driven waves in which amplitudes increase with fetch (Walsh et al., 1989). Another potential contribution to higher speed off shore is a possibly larger concentration of frazil near shore reducing wave height."

p.7 19: Here one should start a new paragraph: " We further compare…"

Done

p. 7 23-24: It is misleading that in the first sentence vgr is used for the drift vector derived from the coastal radar, and in the second sentence for the LOS-component "vgr" derived from vgr.

Now separated and used new terms ($v_g$ and $v_{vp}$)

p. 7 26: Here the averaging time for the coastal radar is given as 2.5 hours, in section 2.1, 1.5 hours were mentioned.

Good point. This is due to the fact that data is not always archived every 5 minutes. This is now explained better in Section 2.1 (P3,L17):

"Radar images are archived roughly every 5-10 minutes"

And (P3,L23):

"The filtering approach uses 18 consecutive images resulting in motion products averaged over 1.5-3 hours with a grid spacing of 430 m. For features trackable over much of the radar image, errors in the dislocation vectors are on average well below 10% (Rohith et al., 2013) where an average 5 % error is attributed to uncertainties related to spatial scale and time intervals (Mahoney et al., 2015b)."

p.8, 1-2: the drift speed error mentioned here is only related to the motion-height ambiguity. But phase noise also contributes to the drift speed error.

Drift speed error as a result of phase noise has now been discussed in Section 2.3 (P5,L16).

p. 8, 2-4: Why is a difference between landfast and adjacent drifting ice of 5 cm not possible? I interpret the SAR amplitude image in Fig. 3 such that there is a mixture of smaller dark appearing (hence thinner) ice floes in open water (bright) - in the middle interrupted by more consolidated thin ice - next to the landfast ice. However, since the image is relatively small in the manuscript, it is difficult to interpret it.

Sea ice is in general not thicker than 0.5 m in Barrow during November. Hence a difference in freeboard is unlikely to be larger than 5 cm of any ice type.

p. 8, 11: the phase cannot be solved accurately, it includes an uncertainty due to phase noise

(which depends on decorrelation effects) and the achievable accuracies in determination of
baselines, incidence angles, co-registration in processing etc. These factors are not mentioned in
section 2 or in the discussion.

This has now been included in Section 2.3.

p. 8, 15-17: "….sea ice and features a…" Isn't it "which features"?

Changed

p. 9, 16-19: This interpretation again raises the question about how representative milliseconds
snapshots are on time-scales of minutes to hours. Convergence should be given in velocity per
length unit. The identification of ridges and rubble fields is relatively difficult at higher frequencies
such as X-band, unless the spatial resolution is high (another reason to mention it in the paper).

This has been included (P10,L7): "roughly 10 cm s$^-$1over a distance of less than 100 m,…"
Resolution has been included in response to previous comment.

p. 9, 29-30: Since only one temporally isolated snapshot of milliseconds LOS velocity is available,
short-lived (or transient) dynamics cannot directly be deduced from the data alone. This requires a
careful additional analysis of the environmental context over a longer time period. Without such an
analysis one does not know whether an event lasts for seconds, minutes or hours.

We have now included this in the conclusions (P13,L3): "Even with the potential application of S-ATI for
evaluating short-lived processes, it will inevitably require a careful analysis of the environmental forcing
over a longer time period. Only then it is possible to know whether the phase-derived drift can be
representative on scales from minutes to hours. In this context, it would also be beneficial to design
experiments to study ice motion on temporal scales from sub-seconds (i.e. S-ATI) to minutes (e.g. coastal
radar, buoys) to several hours (e.g. buoys, SAR) to better understand differences between observations at
different scales and relate standard SAR approaches (e.g. maximum cross correlation) to S-ATI.
Furthermore, an important question is what temporal resolution is required to investigate "short-lived"
events such as transient convergence or strain response upon ice impact with structures. This may also be
detectable on scales of several seconds, but questionable on longer timescales of minutes. It is presently
largely an open question what short-lived processes occur at different timescales and what temporal
baselines would be best suited to capture them."

It also needs to
be clarified what "short-lived" means in terms of time units. Certainly not milliseconds.

This has been clarified (P10,L8): "This suggests the event had only just commenced and / or only
occurred on a timescale of a few seconds."

Section 4
p. 10, 1-2: "high accuracy" – the plot Fig. 3 d does not support this statement, since 2.5 hours
average velocities are compared with milliseconds ice movements. It looks like a "reasonable"
agreement between both velocities, but with maximum deviations up to 0.3-0.4 m/s even when
excluding the young ice values. It would be helpful to get examples of the relative error for different
velocity ranges.

We have now separated the analysis in Section 3.1 into three clusters in Figure 3d. It is clear that the
accuracy is only good (roughly within 0.1 m s$^{-1}$ for floes in free drift. We have commented on that further

and included now in the discussion (P10,L25): "We have demonstrated the potential use of S-ATI for derivation of instantaneous sea ice drift. The phase-derived speed has shown to conform well with ground-radar validation dataset with accuracy within roughly 0.1 m s-1 for ice floes in free drift."

p. 10, first paragraph: the limitations of S-ATI have to be accounted for in this discussion!
• Examples: Opening and closing rates for leads can only be provided for certain orientations between lead and LOS, and information on those rates is only available for the past – for navigation such information may be useful - but as forecast.

The discussion section has now been completely rewritten. We took out references to forecasts and have further discussed limitations. The orientation aspect is included in the conclusions where we now introduce possibilities with future systems.

• Is a millisecond snapshot well suited to separate landfast and temporally stationary ice – and would this really improve ice charting? The duration to classify stationary ice as landfast is often fixed at 20 days. What is then "temporally stationary"? How can a practical approach to distinguish landfast and temporally stationary ice by means of S-ATI look like?

Good point. We have taken out the discussion around ice charting and included other possible applications in the discussion.

p. 10, second paragraph: a reconstruction of the 2D instantaneous drift field from the 1D-LOS component will not be very reliable but the consideration of wind, current and general ice drift information will help to decide when conditions for the application of S-ATI are close to optimal.

We have included a discussion about this point (P11,L10): "However, considering wind, currents and general drift information will help decide when conditions for application of S-ATI are close to optimal."

p. 11, first paragraph: I regard this example of deriving absolute speed from a moving vessel and relative speed between vessel and surrounding ice as an interesting scientific exercise but not useful for ice navigation. The example of protecting offshore installations should be elaborated but I presume that it is more a question of collecting statistics of maximum pressure loads on planned installations instead of giving sporadic information to already existing installations.

We agree. We have now changed this according to your suggestion and moved it to the conclusions (P12,L28): "With the m-scale resolution of stripmap X-band SAR, this approach would likely be able to provide statistics of maximum pressure loads on structures relevant for engineering design and planning of offshore installations."

p.11, third paragraph: No, the S-ATI approach cannot be more accurate than the traditional approaches because of the reasons mentioned above. They are two very different, complementary approaches. It would be worth to design experiments to study ice motion on temporal scales from sub-seconds (S-ATI) to minutes (coastal radar, buoys) to several hours (buoys, SAR constellations) to better relate both approaches to one another.

This claim has been removed and we have included your suggestion in the conclusions (P13,L5): "In this context, it would also be beneficial to design experiments to study ice motion on temporal scales from sub-seconds (i.e. S-ATI) to minutes (e.g. coastal radar, buoys) to several hours (e.g. buoys, SAR) to better understand differences between observations at different scales and relate both standard SAR approaches to S-ATI."

Conclusions:
Comments given above should be considered.

We have followed all your recommendations

p. 12, 19-20: Given (1) the (yet) missing link between millisecond and hours timescales, (2) the difficult interpretation related to the missing information on drift direction, and (3) the lack of tandem-missions useful for operational ice charting it is NOT clear that S-ATI is relevant as a tactical tool for Arctic stakeholders.

The claim that this could be used for tactical decision making has been removed.

Recommendations:
It is not up to the reviewer to demand changes of the paper contents. Nevertheless, I would like to recommend some items for consideration in the discussion:
• It would be useful to compare the approach of Kræmer et al. and the S-ATI technique in some detail since they have similar limitations concerning certain aspects but are also different in other aspects. Move lines 19-23 on p. 2 into the discussion.

We have here taken an alternative approach to your recommendation, namely to not go into too much detail regarding the Kræmer study. In that way it no longer appears that we claim that ATI is superior. We rather focus on the application and limitations of the ATI approach.

• In the Dierking et al. paper the goal was to retrieve sea ice topography, and the influence of ice motion on the interferometric phase was a disturbing factor that needs to be minimized in this type of application. Here, it is vice versa: motion as wanted information and topography as disturbing factor. Which along- and across-track baselines would you recommend, considering the typical ranges of ice drift speed and height elevations?

We have now included more discussion around this (P11,L31): "Small effective baselines will be beneficial due to reduced sensitivity to topography. By combining Equation 3 and 6 by setting $\sigma_v = v_e$ it is possible to determine the maximum $B_\perp$ that will ensure that a given height, $h_0$, will result in a velocity error comparable to that of phase noise. Assuming $\theta = 30°$, and $\sigma_\phi$ similar to what is used in our work based on $\gamma = 0.8$, we determine maximum $B_\perp \sim 2$ km for a large $h_0 = 0.3$ m. A small $v_a$ is also beneficial by reducing sensitivity to noise and $v_e$. As $v_a$ is inversely proportional to the along-track baseline, a possible tenfold increase in $B_\parallel$ leading to $B_t \sim 1$ s would likely be more optimal. A further increase would likely result in increase in rotation of floes and possible deformation, further complicating the results."

• What is the highest temporal resolution that is required for investigating "short-lived" events? Examples of such events? I am sure that we can extrapolate the milliseconds measurements into the range of seconds, but what are the highest possible frequencies of motion changes, and to which events are they linked? This item is very tricky to answer, and it may be sufficient at this time just to raise these questions to make the reader aware of still existing problems.

This has been included in the conclusions (P13,L8): "Furthermore, an important question is what temporal resolution is required to investigate "short-lived" events such as transient convergence or strain response upon ice impact with structures. This may be also be detectable on scales of several seconds, but questionable on longer timescales of minutes. It is presently largely an open question what short-lived processes occur at different timescales and what temporal baselines would be best suited to capture them."

• Since the interpretation of drift fields derived from S-ATI is not straightforward, it may be useful for the reader to get a "recipe" of important factors to be considered (e .g. uncertainty of the measurement which depends on phase noise and along-track baseline, LOS – where to get information of the main drift direction, millisecond snapshot – what has to be expected for the next minutes/hours etc).

We are open to this. However, by including all your suggestions, we think the manuscript and the application of this approach is now much clearer, hence such a "recipe" may not be needed. We are slightly hesitant to include a "recipe", since it may include guesswork and added explanations and discussion, which could make the manuscript more difficult to follow.

[revised manuscript text omitted]

---

## Referee Report (RR1)

Review of „Instantaneous sea ice drift speed from TanDEM-X interferometry"
by D. O. Dammann and 7 co-authors, revised manuscript

General assessment: The major concern of my first review was the authors' judgment regarding the benefit of retrieving sea ice drift from S-ATI data for different applications such as marine operations and safety in sea-ice covered waters or sea ice science. The authors addressed my criticism in depth. The result is that it is now much easier for the reader to understand pros and cons of this technique. Hence I recommend the paper for publication but propose a few minor modifications.

Abstract: lines 14-15 "…S-ATI as a tool to assess ice drift, inherent limitations, and possible applications": this can also be read as "S-ATI as a tool to assess inherent limitations and possible applications". Should be rephrased.

Page 2, line 10: the Muckenhuber and Sandven reference is out of place here, the application of SAR for retrieving ice drift is described in many other papers as well – I think a reference is not needed here

Page 2, line 16: I propose to add two useful papers here:
A. A. Korosov and P. Rampal, A combination of feature tracking and pattern matching with optimal paramtrization for sea ice drift retrieval from SAR data, Remote Sensing 9, 258, 2017, doi:10.3390/rs9030258
T. Hollands, W. Dierking, "Performance of a multi-scale correlation algorithm for the estimation of sea ice drift from SAR images: initial results", *Annals of Glaciology* 52(57), pp. 311-317, 2011

Page 2, line 18: the error of the retrieved ice drift does not only depend on the complexity of the ice drift patterns but also on the spatial resolution of the SAR images and the temporal gap between the image pair from which ice displacements are retrieved. However, Hutchings et al. did not use SAR images for their analysis, and I guess this is also valid for the Haller paper? Hence the hints to those studies are misleading here.

Page 2, line 21: "Instantaneous drift estimates…supplement…traditional SAR-based ice drift algorithms for improved accuracy". I agree that both methods can supplement each other, but I don't see how S-ATI results can be used directly to improve the accuracy of the traditional methods.

Page 2, line 31: another useful paper regarding retrieval of sea ice topography on landfast ice is:
T. G. Yitayew , W Dierking, D. V. Divine, T. Eltoft, L. Ferro-Famil, A. Rösel, and J. Negrel, "Validation of sea-ice topographic heights derived from TanDEM-X interferometric SAR data with results from laser profiler and photogrammetry , *IEEE Trans. Geosci. Remote Sens.*, vol. 56, no. 11, pp. 6504-6520, 2018, doi:10.1109/TGRS.2018.2839590

Page 3, lines 25-26 "errors in the dislocation vectors" - do the given numbers refer to the magnitude of the vector? Or are they valid both for magnitude and direction (orientation)?

Page 4, line 2: "accuracy of BETTER than 0.01 ms$^{-1}$"?

Page 5, line 9 (above equation 2): since your displacement of ambiguity is given for ground range, you should make clear that also your $v_\phi$ is the *ground* speed in look-direction (most equations in the literature give the line-of-sight velocity since the vertical component of ground movement is not always zero).

Page 5, line 22: "sections of open water": the speed of Bragg waves is sensitive to short-scale wind changes, hence the motion of the sea surface may reveal larger variations, see e.g. the lead in Fig 4a in the Dierking et al 2017 TC paper (where motion is interpreted as height).

Page 6, line 4 "…the average phase of the flow can be used to describe the linear motion". Actually this is not valid. Even after phase unwrapping the phase is ambiguous, and ice floes are not necessarily symmetric, which is one assumption in the Scheiber-paper. Did you observe indications of rotation in your data? If not, you should mention it.

Page 7, lines 31-32: Rephrasing: "Another potential reason for an increasing speed with increasing distance from the shore would be a larger concentration of…"

Page 8, line 15: there is one $v_\phi$ too much.

Page 9, line 2 "…phase can be used to accurately derive ice…"?

Page 11, lines 21-28: One should mention that the consideration of the topographic phase is necessary here because the perpendicular baseline is > 0.

Page 11, line 28: In Fig. 4a there seem to be no indications of icebergs at positions of bright spots that occur in Fig. 4b. Are there other explanations for the outliers?

Page 11, lines 32-33, page 12 lines 1-2: The first sentence is somewhat oddly phrased. Suggestion: "Hence the question is how large the acceptable length of the normal baseline $B_n$ is so that the effect of topography (quantified by the maximum height) corresponds to the effect of phase noise on the phase derived speed. By combining equations (3) and (6) and setting $v_e = \sigma_v$ we can determine the baseline $B_n$ for a given height $h_0$. Assuming…" or something similar…

---

## Author Response (AR2)

Review of „Instantaneous sea ice drift speed from TanDEM-X interferometry"
by D. O. Dammann and 7 co-authors, revised manuscript

General assessment: The major concern of my first review was the authors' judgment regarding the benefit of retrieving sea ice drift from S-ATI data for different applications such as marine operations and safety in sea-ice covered waters or sea ice science. The authors addressed my criticism in depth. The result is that it is now much easier for the reader to understand pros and cons of this technique. Hence I recommend the paper for publication but propose a few minor modifications.

Dear Professor Dierking,

Thank you for once again reviewing our manuscript and providing important corrections and comments. This is very appreciated. We have changed the manuscript according to your recommendations leading to an improved paper.

Best regards,
Dyre Dammann

Abstract: lines 14-15 "…S-ATI as a tool to assess ice drift, inherent limitations, and possible applications": this can also be read as "S-ATI as a tool to assess inherent limitations and possible applications". Should be rephrased.

Good point. done

Page 2, line 10: the Muckenhuber and Sandven reference is out of place here, the application of SAR for retrieving ice drift is described in many other papers as well – I think a reference is not needed here

Taken out

Page 2, line 16: I propose to add two useful papers here:
A. A. Korosov and P. Rampal, A combination of feature tracking and pattern matching with optimal paramtrization for sea ice drift retrieval from SAR data, Remote Sensing 9, 258, 2017, doi:10.3390/rs9030258
T. Hollands, W. Dierking, "Performance of a multi-scale correlation algorithm for the estimation of sea ice drift from SAR images: initial results", Annals of Glaciology 52(57), pp. 311-317, 2011

Done

Page 2, line 18: the error of the retrieved ice drift does not only depend on the complexity of the ice drift patterns but also on the spatial resolution of the SAR images and the temporal gap between the image pair from which ice displacements are retrieved. However, Hutchings et al. did not use SAR images for their analysis, and I guess this is also valid for the Haller paper? Hence the hints to those studies are misleading here.

Agree. Rephrased: "The derivation of sea ice drift speed from buoy data sampled every 1-3 days has been found to lead to underestimation of ice drift speeds by 10-20% (Haller et al., 2014) but can likely be much higher (Hutchings et al., 2011). The same bias can thus be expected when applying SAR with similar temporal sampling on the order of days."

Page 2, line 21: "Instantaneous drift estimates…supplement…traditional SAR-based ice drift algorithms for improved accuracy". I agree that both methods can supplement each other, but I don't see how S-ATI results can be used directly to improve the accuracy of the traditional methods.

Taken out

Page 2, line 31: another useful paper regarding retrieval of sea ice topography on landfast ice is: T. G. Yitayew , W Dierking, D. V. Divine, T. Eltoft, L. Ferro-Famil, A. Rösel, and J. Negrel, "Validation of sea-ice topographic heights derived from TanDEM-X interferometric SAR data with results from laser profiler and photogrammetry , IEEE Trans. Geosci. Remote Sens., vol. 56, no. 11, pp. 6504-6520, 2018, doi:10.1109/TGRS.2018.2839590

Included

Page 3, lines 25-26 "errors in the dislocation vectors" - do the given numbers refer to the magnitude of the vector? Or are they valid both for magnitude and direction (orientation)?

Yes, this refers to the magnitude. The mistracked points are expected to be randomly distributed around the true position for the types of signals/features tracked by the radar.

Page 4, line 2: "accuracy of BETTER than 0.01 ms$_{-1}$"?

Yes. corrected

Page 5, line 9 (above equation 2): since your displacement of ambiguity is given for ground range, you should make clear that also your v$_\phi$ is the ground speed in look-direction (most equations in the literature give the line-of-sight velocity since the vertical component of ground movement is not always zero).

Good point. Done

Page 5, line 22: "sections of open water": the speed of Bragg waves is sensitive to short-scale wind changes, hence the motion of the sea surface may reveal larger variations, see e.g. the lead in Fig 4a in the Dierking et al 2017 TC paper (where motion is interpreted as height).

Good point. Thank you for pointing this out. We realize here that even mentioning open water may be unnecessary as we are focusing on deriving ice speed. Therefore, we took out the mentioning of open water here all together.

Page 6, line 4 "…the average phase of the flow can be used to describe the linear motion". Actually this is not valid. Even after phase unwrapping the phase is ambiguous, and ice floes are not necessarily symmetric, which is one assumption in the Scheiber-paper. Did you observe indications of rotation in your data? If not, you should mention it.

Good point. This has been taken out. We also now mention that we did not see indications of rotation in our case studies.

Page 7, lines 31-32: Rephrasing: "Another potential reason for an increasing speed with increasing

distance from the shore would be a larger concentration of…"

Changed

Page 8, line 15: there is one $v_\phi$ too much.

Taken out

Page 9, line 2 "…phase can be used to accurately derive ice…"?

Done

Page 11, lines 21-28: One should mention that the consideration of the topographic phase is necessary here because the perpendicular baseline is > 0.

Done

Page 11, line 28: In Fig. 4a there seem to be no indications of icebergs at positions of bright spots that occur in Fig. 4b. Are there other explanations for the outliers?

The phase signatures of these areas correspond to heights of tens of meters similar to expected iceberg heights in this region. As we can see, there is no other explanation for this signature. In fact, this observation, that icebergs can be detected and evaluated with InSAR have led to an entirely new manuscript presently also in review in The Cryosphere, which we now are citing:

Dammann, D.O.; Eriksson, L.E.B.; Nghiem, S.V.; Pettit, E.; Kurtz, N.T.; Sonntag, J.; Busche, T.; Meyer, F.; Mahoney, A. Iceberg topography and volume classification using TanDEM-X interferometry. The Cryosphere, In review.

Page 11, lines 32-33, page 12 lines 1-2: The first sentence is somewhat oddly phrased. Suggestion: "Hence the question is how large the acceptable length of the normal baseline $B_n$ is so that the effect of topography (quantified by the maximum height) corresponds to the effect of phase noise on the phase derived speed. By combining equations (3) and (6) and setting $v_e = \sigma_v$ we can determine the baseline $B_n$ for a given height $h_0$. Assuming…" or something similar…

Great suggestion. Done

[revised manuscript text omitted]

---

## Author Response (AR3)

Dear Dr. Kaleschke,

Thank you for handling our manuscript and providing great feedback. We have now included author contributions. We have also attached the motion vectors as supplementary data. We greatly appreciate the input on our validation. We agree that your suggestion would enable us to test the ATI technique with more data. However, we think what we have done so far is sufficient for a first demonstration and we would prefer to keep the paper and the results presented in it as they are and to publish them in this form. The inclusion of additional data would not only require more work on our side, but it could be considered as such a significant modification that a complete new round of reviews could be required.

Thank you again for your assistance with our manuscript.

Best regards,
Dyre Dammann